# Exponential Moving Average of Weights in Deep Learning: Dynamics and Benefits

**Daniel Morales-Brotons**                                    *danimoralesbrotons@gmail.com*
*EPFL*

**Thijs Vogels**
*EPFL*

**Hadrien Hendrikx**                                          *hadrien.hendrikx@inria.fr*
*Centre Inria de l'Univ. Grenoble Alpes, CNRS, LJK, Grenoble, France*

**Reviewed on OpenReview:** *https://openreview.net/forum?id=2M9CUnYnBA*

## Abstract

Weight averaging of Stochastic Gradient Descent (SGD) iterates is a popular method for training deep learning models. While it is often used as part of complex training pipelines to improve generalization or serve as a 'teacher' model, weight averaging lacks proper evaluation on its own. In this work, we present a systematic study of the Exponential Moving Average (EMA) of weights. We first explore the training dynamics of EMA, give guidelines for hyperparameter tuning, and highlight its good early performance, partly explaining its success as a teacher. We also observe that EMA requires less learning rate decay compared to SGD since averaging naturally reduces noise, introducing a form of implicit regularization. Through extensive experiments, we show that EMA solutions differ from last-iterate solutions. EMA models not only generalize better but also exhibit improved i) robustness to noisy labels, ii) prediction consistency, iii) calibration and iv) transfer learning. Therefore, we suggest that an EMA of weights is a simple yet effective plug-in to improve the performance of deep learning models.

## 1 Introduction

The performance of modern deep learning models is tightly linked to their training. In order to converge to a good solution, reducing the noise coming from stochastic updates is eventually required. For example, Stochastic Gradient Descent (SGD) (Robbins & Monro, 1951; Bottou, 2010) needs a carefully tuned learning rate and decay schedule, while adaptive variants such as Adam (Kingma & Ba, 2014) essentially decay the learning rate based on gradients retrieved: while a large learning rate is initially required to obtain fast training and good generalization, only when the learning rate is low enough does the model actually converge to a good solution. In an orthogonal approach to learning rate decay, a standard way to reduce noise in *convex* optimization is (tail) averaging (Polyak & Juditsky, 1992): if SGD stops making progress because of stochastic noise, a more accurate solution can be retrieved despite high learning rates by averaging the last iterates.

While the theoretical gains of averaging are less clear in the non-convex setting, weight averaging (WA) is also popular in deep learning, and has been explored primarily in two ways: WA inside the training loop as a teacher model, and WA outside the training loop to improve generalization. In the first case, an Exponential Moving Average (EMA) of parameters is used as a teacher in Student-Teacher frameworks, for example in popular representation learning methods (Tarvainen & Valpola, 2017; Grill et al., 2020). The averaged model provides more accurate and consistent predictions during training, which the student uses for training. For the second case, Stochastic Weight Averaging (SWA) (Izmailov et al., 2018) uses an average of multiple

checkpoints along the SGD trajectory to improve the generalization of the final model, arguing that it finds flatter solutions than SGD. Note that SWA does not affect optimization since the averaged model is not used in the training loop.

Despite the popularity of EMA teachers, the properties of EMA models have not been studied thoroughly. Previous works that use EMA (Grill et al., 2020; He et al., 2020) mainly justify its effectiveness by enhanced consistency and stability of predictions during training, and often mix EMA with other mechanisms as part of a complex framework, making it impossible to disentangle the impact of averaging. EMA has mainly been used as a teacher, leaving aside the potential of EMA itself to improve the final solution in favor of SWA.

In this work, we focus on EMA models outside of the training loop, exploring their training dynamics and benefits in generalization and beyond. By doing so, we unveil new reasons why EMA are such good teachers: we find that the solutions reached when reducing stochastic noise by averaging are different than by learning rate decay. They improve in robustness to label noise, calibration, prediction consistency and transfer learning. In a nutshell, we ask the following question:

**What are the properties of weight averaging when training deep neural networks?**

To answer this question, we empirically study weight averaging during a (momentum) SGD trajectory by means of an Exponential Moving Average (EMA). More specifically, if the SGD models form a trajectory $(\mathbf{x}_t)_{t \geq 0}$, the corresponding EMA models would be obtained by taking, for some $\alpha \in [0, 1]$:

$$\mathbf{x}_0^{\text{EMA}} = \mathbf{x}_0, \text{ and } \mathbf{x}_{t+1}^{\text{EMA}} = \alpha \, \mathbf{x}_t^{\text{EMA}} + (1 - \alpha) \, \mathbf{x}_{t+1}. \tag{1}$$

Keeping an EMA model is simple, has minimal overhead, and can easily be plugged into any existing pipeline. We study EMA models outside of the training loop, such that they have no effect on the underlying SGD trajectory. We split our study in two parts, described in the two paragraphs below.

**Training dynamics of EMA (Sec. 3).** We find that weight averaging reduces noise in the model parameters compared to SGD and can replace the last phase of learning rate decay, while at the same time enabling implicit regularization via stochastic noise. More specifically, **(i)** While last-iterate SGD requires decaying the learning rate to nearly 0 for convergence, averaging reduces noise and yields good solutions under reasonably high learning rates, which we argue promotes learning more general representations. We propose a one-shot tuning of the strength of implicit regularization by using cosine annealing of the learning rate and early stopping. **(ii)** We highlight the impressive performance of EMA in the early stages of training and postulate this observation as a key reason for the success of popular EMA teachers. Combined with early stopping, the EMA model can reduce the compute budget by sparing the last phase of SGD training at low learning rates. **(iii)** We look into Batch Normalization (BN) in EMA models and find that it is the limiting factor when choosing the EMA decay. If BN statistics are recomputed, larger averaging windows can be used and actually may improve further generalization.

**Properties of the final EMA model (Sec. 4).** We find that the solutions reached with EMA and early stopping are different from the baseline solution obtained by last-iterate momentum SGD, and conjecture that this is due to implicit regularization via stochastic noise, obtained through the use of larger step-sizes. More specifically, **(i)** in extensive experiments on image classification tasks we find a consistent improvement in **generalization** using an EMA model. **(ii)** We also find a great improvement in **robustness to label noise** in training data, as the implicit regularization largely prevents memorization of wrong labels. A simple EMA model proves to be competitive with specialized methods for robust training to label noise. **(iii)** We compare the EMA model to the SGD baseline in a number of other metrics and find that it improves considerably in **calibration**, **prediction consistency** and **transfer learning**.

## 2 Related Work

Weight averaging during deep learning training is already a popular method that leads to better practical performances. Yet, it lacks a systematic study, since contributions are mostly scattered in domain-specific literature. Besides, the use of weight averaging is often justified by alleged intuitive properties which are rarely investigated in isolation. In this section, we review different areas in which weight averaging is used, along

with the corresponding alleged folklore benefits. We then test them, as well as give alternative explanations for why weight averaging is useful in the remaining sections.

**Weight averaging to improve generalization.** Averaging the iterates during a trajectory has a long history in stochastic approximation (Ruppert, 1988; Polyak, 1990; Polyak & Juditsky, 1992), and its correct use and understanding have been an active area of research (Bach & Moulines, 2011; Dieuleveut et al., 2017; Lakshminarayanan & Szepesvari, 2018; Mücke et al., 2019; Gadat & Panloup, 2023). Geometric averaging (EMA but with more weights on old iterates) can also be connected with a form of explicit regularization Neu & Rosasco (2018). Yet, these methods assume quadratic or (strongly) convex objectives, and so do not apply to deep learning training.

In deep learning, a popular averaging method is Stochastic Weight Averaging (SWA) (Izmailov et al., 2018). SWA keeps a uniform average of checkpoints during the final epochs of an SGD trajectory, while holding a reasonably high and constant learning rate. SWA is argued to find flatter solutions than SGD, thus generalizing better to unseen data. A potential explanation is that the loss function near a minimum is often asymmetric, sharp in some directions and flat in others. While SGD tends to land near a sharp ascent, averaging iterates biases solutions towards a flat region (He et al., 2019).

Many extensions of SWA have been proposed for specialized tasks (Gupta et al., 2020; Li et al., 2022; Kaddour, 2022), and in particular semi-supervised learning (Athiwaratkun et al., 2019), low-precision training (Yang et al., 2019) and domain generalization (Cha et al., 2021). The latter introduces enhancements such as dense averaging (*i.e.*, every iteration) and overfit-aware sampling by tracking validation loss. The flatness argument has also been leveraged for robustness: weight averaging on top of adversarial training helps finding flatter minima and boosts adversarial robustness. This has been shown using both SWA (Chen et al., 2021) and EMA (Gowal et al., 2020; Rebuffi et al., 2021). EMA has also been studied in minimax optimization, with applications to GANs (Yaz et al., 2019). Furthermore, many works use averaging as part of their implementation but do not emphasize it or discuss its effect. Such works are hard to review since they are not explicitly listed as working on averaging, but include for instance Berthelot et al. (2019); Sohn et al. (2020); Oord et al. (2018); Oquab et al. (2023), which rely on EMA or uniform averaging, sometimes replacing the decay of learning rate. Sanyal et al. (2023) use weight averaging to improve results of LLMs pre-training. Finally, Sandler et al. (2023) provide an analytical model the behavior of the high-dimensional vector of parameters along an SGD trajectory, showing an improvement in generalization and finding an equivalence between averaging and learning rate decay.

**Weight averaging in Student-Teacher methods.** Consistency training is a popular technique for learning with unlabeled data (Laine & Aila, 2016; Berthelot et al., 2019), based on generating pseudo-labels during training, often through a *teacher* model, which does not receive gradient updates. Mean Teacher (Tarvainen & Valpola, 2017) first proposed to use an EMA of model weights as a teacher, such that $\theta' = \text{EMA}(\theta)$, in a method for semi-supervised image classification. EMA has since become a popular choice for teacher models, used for tasks such as semi-supervised semantic segmentation (French et al., 2019), unsupervised domain adaptation (Hoyer et al., 2022), continual adaptation (Wang et al., 2022), and robustness to label noise Liu et al. (2020); Nguyen et al. (2019). On the other hand, SWA requires recomputing batch norm (BN) statistics for the averaged model with a full pass over the train set, thus making it unfit for its online use (*e.g.*, as teacher). While these works find that EMA teachers are beneficial and provide accurate pseudo-labels, they do not specifically study the properties of EMA models.

In self-supervised learning, EMA plays a central role in a handful of popular frameworks. BYOL (Grill et al., 2020) employs consistency training with an EMA teacher (*a.k.a.* self-distillation) to learn visual representations from unsupervised data. MoCo (He et al., 2020) rebrands the EMA teacher as a momentum encoder and proposes a student-teacher framework with a contrastive learning objective. CURL (Laskin et al., 2020) applies the same idea to learn unsupervised representations for reinforcement learning. These methods attribute the effectiveness of EMA to smoother changes in target representations, maintaining consistency and stability, rather than the quality of the representations. DINO (Caron et al., 2021) explores self-distillation in Transformers and studies, for the first time, the training dynamics of the EMA teacher, including the key observation that the teacher consistently outperforms the student during training.

In summary, weight averaging is a key component of student-teacher methods. EMA is generally preferred over other averaging methods (such as SWA) to avoid recomputing BN stats. In this work, we investigate the relation between averaging window and BN statistics, and show that this is actually only the case for short averaging windows.

## 3 Insights on Weight Averaging during Training

Although EMA models are built from SGD iterates, their dynamics during training and final solutions are very different. We argue that EMA is a simple, lightweight and effective plug-in to SGD training.

### 3.1 Training with EMA

**Computation overhead.** The overhead of using an EMA of weights outside of the training loop is generally very low, as it only requires keeping a running average of parameters and possibly evaluating every epoch. Moreover, the running average can be updated every $T$ steps instead of after every parameter update. We set $T = 16$ by default and find no difference to $T = 1$ in the results. In terms of computation, the optimization step remains the dominant factor by orders of magnitude. In terms of memory, keeping an additional set of weights is feasible for most deep learning models used in practice, other than foundation models. For example, a ResNet-50 (23.7M parameters) requires 90.43 MB of storage.

**Hyperparameters tuning.** There are two main sources of potential tuning overhead when training averaging models: 1) deciding on the averaging window and 2) tuning the final learning rate, a sensible hyperparameter crucial for the final performance. The averaging window for an EMA is determined by the decay factor $\alpha$. An EMA naturally avoids the need for (1), since we can simultaneously keep multiple EMA models with different decays to compare different averaging windows. We prevent (2) by using cosine annealing of the learning rate and finding the best early stopping epoch on a validation set. This allows us to search for (1) and (2) on the go, training only once. Admittedly, keeping multiple EMAs (say, $M$) to avoid tuning does increase the overhead by a factor $M$, but it is still a tiny fraction of the computation time for small enough values (*e.g.*, $M = 5$). With this scheme, we need to decide on epoch budget, number and selection of EMA decays, and to search for the best initial learning rate, as usual for regular training of DNNs.

### 3.2 Implicit Regularization with SGD Noise and Learning Rate Schedule

Noisy SGD updates are argued to bias solutions towards flatter regions that are believed to generalize better, partly explaining the success of deep learning (Keskar et al., 2016). This implicit regularization effect makes a case for large learning rates and small mini-batches (Pesme et al., 2021; Even et al., 2023). Nonetheless, standard training of DNNs requires decaying the learning rate to reduce stochastic noise and converge to a good solution. **Averaging during training is an alternative way of to reduce noise in SGD iterates** and reach a good solution without too much learning rate decay. This allows to freely tune the final learning rate to control the strength of implicit regularization, while still converging to a good solution within the neighborhood.

We demonstrate the dynamics of EMA models in Fig. 1a, where we use a continuous decaying of the learning rate $\eta$ with cosine annealing and track the validation accuracy of the EMA model to find the best early stopping epoch (for implementation details see Sec. 4.1). The EMA accuracy of Fig. 1a is the maximum among the 5 EMA models with different decays, plotted in Fig. 1b, which is dominated by $\alpha = 0.998$ at the maximum. We first highlight that the EMA model outperforms SGD throughout training. We also observe that the best EMA model is obtained when averaging updates with a reasonably high learning rate, benefiting from a stronger implicit regularization. The EMA accuracy rises fast at first, then slowly increases as $\eta$ is continuously decayed (and so does the strength of regularization), peaks at epoch 150, and finally deteriorates as $\eta$ is decayed further. Early stopping while sweeping through the learning rate values allows for a one-shot tuning of the regularization strength. Finally, the EMA model matches the SGD sequence when the iterates don't advance ($\eta \to 0$). For more examples of EMA dynamics, see Fig. 2 and App. A. As we will see in Sec. 4, the solutions reached by EMA and SGD are different: the implicit regularization of averaging improves generalization and promotes learning more general representations.

We emphasize that the EMA solutions not only generalize similarly or better than SGD, but also require fewer training epochs. In our experiments with cosine annealing, early stopping for EMA was always at $< 3/4$ of the epochs budget (see App. B). This suggests that the last phase of SGD training is *mostly wasteful*, as the iterates are already around a good solution that cannot be accessed (without averaging) because of stochastic noise. With averaging, there is no need to decay the learning rate that much, and the entire last phase of training can be spared.

### 3.3   EMA in early training

The noise reduction of weight averaging does not only improve the generalization of the final model, but all throughout training. A key difference in the training dynamics of EMA and SGD models is the early-stage performance: **EMA models are very effective early in training**, as shown in Fig. 1a. Its remarkable performance after just a few epochs partly explains the success of EMA teachers. While popular self-supervised methods (Grill et al., 2020; He et al., 2020) attribute the benefit of a slow-moving average to an improved consistency between predictions during training, we argue that the improved quality of the EMA representations also plays a crucial role in their frameworks. Thanks to noise reduction from averaging, the EMA model can achieve notable performances while keeping a large learning rate for fast progress. Student-teacher frameworks leverage this fact and distill knowledge from the EMA teacher.

Instead of knowledge distillation, a tempting idea is to regularly bootstrap the SGD model with EMA parameters. We investigate this (see App. C), but conclude that *bootstrapping with the EMA does not offer any benefit.* The EMA model is simply a good point within the local neighborhood of the latest iterates. After bootstrapping, noisy SGD updates quickly take over and deteriorate the model performance. Therefore, *distillation methods are a more effective way of leveraging EMA's early performance to improve training.*

### 3.4   Batch Norm Statistics and EMA decay

Batch Normalization (BN) presents a challenge for weight averaging. By default, the EMA model uses BN statistics (mean and standard deviation of each activation) from the current batch, but the cross-sample dependency may harm generalization. Cai et al. (2021) improve EMA teachers by keeping a moving average of the BN statistics of the student, which we use in our implementation. SWA (Izmailov et al., 2018) on the other hand recomputes BN statistics for the final averaged model with an additional full pass over the train

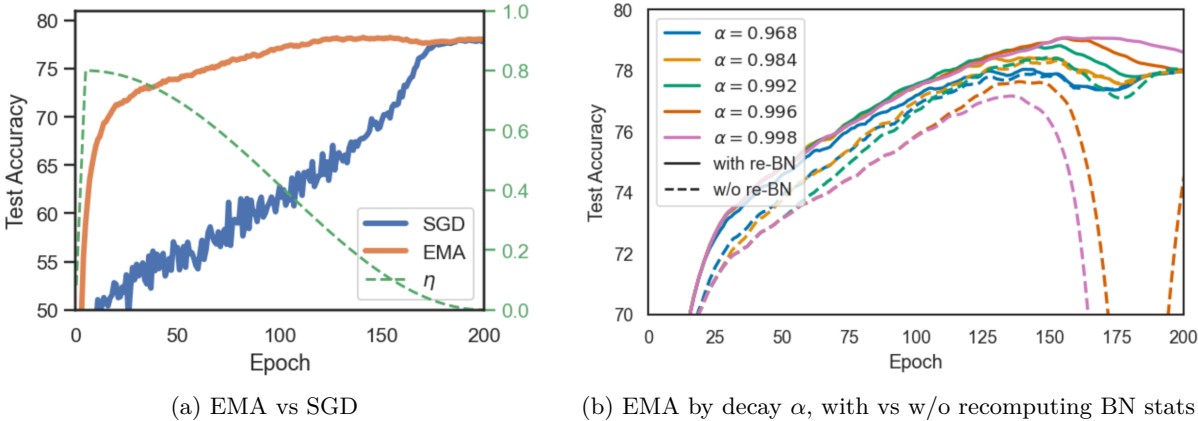

(a) EMA vs SGD          (b) EMA by decay $\alpha$, with vs w/o recomputing BN stats

Figure 1: CIFAR-100 on ResNet-18. **Left:** EMA vs SGD baseline, and learning rate ($\eta$). EMA is the best among the 5 EMA models at any given epoch, without recomputing BN stats (*i.e.* the maximum among EMA models plotted on 1b). We observe that EMA dominates momentum SGD and has a good performance since early on. EMA peaks at epoch 150, at the *optimal $\eta$*, and then deteriorates. **Right:** Breakdown of the 5 EMA models per decay (with and without BN recomputation after every epoch). EMAs with the largest averaging windows fail unless BN stats are recomputed. Sliding window of 5 used for smoothing. All results are the mean of 3 runs.

data after training. For the online use of an EMA model, however, recomputing BN stats during training would imply a significant overhead.

We investigate the optimal averaging window size (*i.e.*, decay $\alpha$) for EMA and find different behaviors for the running average of model parameters and of BN stats. In particular, **model parameters tolerate larger averaging window sizes than BN statistics**. As shown in Fig. 1b, the EMA model can *diverge* when a very slow decay (*i.e.*, large $\alpha$) is used. Interestingly, if we recompute the BN stats of that same EMA model (once after every epoch) we recover full performance, indicating that it is BN that breaks an EMA model as we increase the size of the averaging window. In fact, if recomputing BN stats, averaging weights tends to benefit from using very large averaging windows. We also observe that recomputing BN stats always improves generalization.

The EMA decay is usually set to $\alpha \in [0.9, 0.9999]$, and the optimal value will be task dependent. For an online use of EMA, when recomputing BN stats periodically may be undesirable, a faster decay may be used to avoid divergence. On the other hand, when averaging to improve final performance, it is preferable to use a slower decay and recompute BN stats after training. Models that do not use BN (*e.g.*, VGG-16, Transformer models) naturally avoid this problem.

## 4 Results

### 4.1 Experimental Setup

We perform experiments on several image classification datasets (CIFAR-10, CIFAR-100, Tiny-ImageNet (Le & Yang, 2015)) with various network architectures (ResNet-18 (He et al., 2016), WideResNet-28-10 (Zagoruyko & Komodakis, 2016), VGG-16 (Simonyan & Zisserman, 2014)). We always use SGD with Nesterov Momentum of 0.9 for training (Loshchilov & Hutter, 2017). The epochs, batch size and weight decay are fixed (see details in App. F). For the learning rate schedule, we use a linear warmup during the first 5 epochs and then decay with cosine annealing, and search for the best initial learning rate. We always report the mean of 3 independent runs.

To perform a rigorous study, we stress the importance of using a hold-out set for hyperparameter selection. Unfortunately, most image classification benchmarks do not include a standard validation set. We define random 80/20 splits of the training set for train and validation respectively and perform hyperparameter optimization on the validation set, including the early stopping epoch for EMA (without BN stats recomputation). Finally, we train on the full training data using the selected hyperparameters and evaluate on the test set. Note that early stopping is not tuned again on the test set. Also note that most methods would technically require this train/evaluation split: the best step-size for SGD should be selected on a validation set for instance. We explicitly use one here instead of choosing the best performance on the test set (as is often done) since our EMA training pipeline also relies on early stopping, which could be misleading if chosen directly on the test set.

The EMA introduces one hyperparameter, the decay rate $\alpha$, which governs how fast the moving average forgets past iterations. Since the EMA is outside of the training loop, we can optimize $\alpha$ in a single training run by keeping 5 parallel EMA models. We fix the decays to $\alpha \in [0.968, 0.984, 0.992, 0.996, 0.998]$, and use an EMA sampling period of $T = 16$ steps, to reduce the overhead at no cost in performance. Note that using $T > 1$ affects the effective decay, which becomes $\alpha^{1/T}$ (*e.g.*, $0.984^{1/16} \approx 0.999$). We also use a decay warm-up for a faster EMA update in the first epochs, as $\min(\alpha, \frac{t+1}{t+10})$ at time $t$. For EMA's BN statistics we follow Cai et al. (2021).

In our experiments, we compare *Baseline* against *EMA*. We refer as *Baseline* to the momentum SGD model on which we perform the EMA. For the EMA we consider two different early stopping epochs: at *best validation accuracy* and *lowest validation loss*, which are often not aligned and produce solutions with different properties. In both cases, we report the EMA with the largest decay ($\alpha = 0.998$) and recompute BN stats once after training. In App. B we report the full results including the two EMA variants with and without BN recompute.

| Dataset | Architecture | Baseline | | EMA (acc.) | | EMA (loss) | | SWA |
|---|---|---|---|---|---|---|---|---|
| | | Acc. | Loss | Acc. | Loss | Acc. | Loss | Acc. |
| CIFAR-100 | ResNet-18 | 77.63 $\pm$0.14 | 1.02 | 78.55 $\pm$0.28 | 0.84 | 78.07 $\pm$0.29 | 0.82 | **78.69** $\pm$0.25 |
| | VGG-16 | 72.82 $\pm$0.17 | 1.77 | **73.64** $\pm$0.13 | 1.13 | 72.3 $\pm$0.19 | 1.06 | 73.28 $\pm$0.19 |
| | WRN-28-10 | 81.07 $\pm$0.12 | 0.78 | **82.72** $\pm$0.16 | 0.67 | 81.90 $\pm$0.16 | 0.64 | 82.71 $\pm$0.19 |
| CIFAR-10 | ResNet-18 | 95.25 $\pm$0.11 | 0.22 | 95.62 $\pm$0.11 | 0.15 | 95.46 $\pm$0.18 | 0.15 | **95.75** $\pm$0.13 |
| TinyImageNet | ResNet-18 | 66.03 $\pm$0.26 | 1.60 | 67.97 $\pm$0.14 | 1.35 | 67.06 $\pm$0.18 | 1.36 | **68.11** $\pm$0.2 |

Table 1: Test accuracy and loss on a baseline model and its EMA. The EMA model consistently outperforms the baseline in accuracy and loss, and the same is true for SWA. The **best** and second-best accuracies are split between EMA and SWA, with no averaging method performing clearly better. We explore EMA with two early stopping criteria: best accuracy and lowest loss. EMA models' BN statistics are recomputed once.

## 4.2 Generalization

We start by investigating the performance of EMA models in terms of test accuracy. We find that averaging with EMA improves generalization, always outperforming the SGD baseline. This is not unexpected, as the generalization benefit of (uniform) averaging is well-known in deep learning (Izmailov et al., 2018). Nonetheless, to the best of our knowledge, we are the first to show this for EMA.

In Table 1 we report test accuracy and test loss for the momentum SGD baseline and its EMA, early stopped either at the epoch of best accuracy or lowest loss. We emphasize two takeaways. Firstly, EMA performs consistently better than the baseline. We also perform and report SWA experiments, which bring a generalization improvement approximately similar to EMA, with no weight averaging method performing clearly better. Secondly, the EMA model with the lowest loss does not correspond to the model with highest accuracy, as the early stopping point to minimize the loss is always earlier (see App. B). As we discuss in the next sections, the EMA with the lowest loss outperforms the best accuracy EMA in other metrics (*e.g.*, calibration, prediction consistency, transferability), suggesting a trade-off between maximizing these metrics or model accuracy.

## 4.3 Label Noise

In this section, we study the case of training with label noise, *i.e.*, with a fraction of the training set wrongly annotated. We perform experiments on the benchmarks of CIFAR-10N and CIFAR-100N (Wei et al., 2022), two datasets with human annotator label noise of approximately 40%.

Interestingly, we find that the effect of implicit regularization is magnified in the presence of label noise. In Fig. 2 we observe that the EMA model performs best when averaging iterates at a relatively high learning rate. The EMA model peaks at 65.15% accuracy at epoch 100, with a learning rate around 0.4, and then decays until it plateaus at 55.5% at the end of training. Eventually, when the learning rate is decayed low enough, the model fits (*i.e.*, memorizes) all the noisy labels and reaches 100% train accuracy (App. E.1), but generalizes worse. Memorization in the EMA occurs as the learning rate is decayed, and not due to continued training (App. E.2). An explanation for the outstanding performance of EMA is that the regularizing effect of stochastic noise effectively prevents fitting the noisy labels, while it allows learning of general patterns in the data.

We compare our results to the leaderboard for robust training to label noise (see Tab. 2). The performance of the EMA under label noise not only is a good example of the effect of implicit regularization, but it is actually a competitive method with the state-of-the-art, despite its striking simplicity. The leading methods in Tab. 2 are complex specialized frameworks, often computationally demanding. Most of them, including the top-performing DivideMix (Li et al., 2020), train two networks simultaneously while refining labels based on their predictions, and use advanced augmentation strategies such as MixUp. In contrast, we do not adopt any specialized technique or data augmentation, we only keep an EMA model on top of vanilla momentum SGD training. Despite its simplicity, EMA outperforms multiple specialized methods and gets reasonably close to

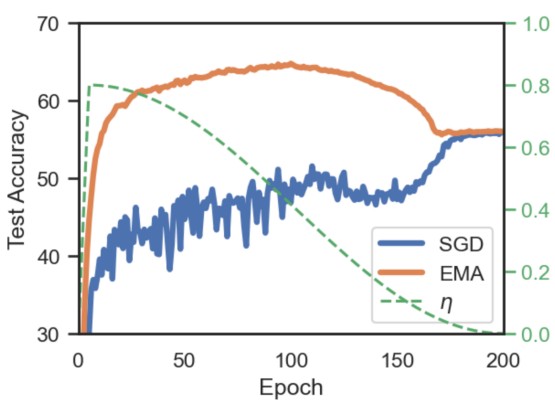

| Method | CIFAR-10N | CIFAR-100N |
|---|---|---|
| DivideMix | 92.56 ±0.42 | 71.13 ±0.48 |
| PES(semi) | 92.68 ±0.22 | 70.36 ±0.33 |
| ELR+ | 91.09 ±1.6 | 66.70 ±0.07 |
| **EMA (acc.)** | **86.71** ±0.17 | **65.15** ±0.20 |
| CAL | 85.36 ±0.16 | 61.73 ±0.42 |
| CORES | 83.60 ±0.53 | 61.15 ±0.73 |
| Co-Teaching | 83.83 ±0.13 | 60.30 ±0.27 |
| JoCor | 83.37 ±0.30 | 59.97 ±0.24 |
| ELR | 83.58 ±1.13 | 58.94 ±0.92 |
| Negative-LS | 82.99 ±0.36 | 58.59 ±0.98 |
| Co-Teaching+ | 83.26 ±0.17 | 57.88 ±0.24 |
| CORES* | 91.66 ±0.09 | 55.72 ±0.42 |
| . . . | . . . | . . . |
| CE (standard) | 77.69 ±1.55 | 55.50 ±0.66 |

Figure 2: CIFAR-100N on ResNet-34. EMA vs SGD baseline, and learning rate $\eta$. EMA dominates SGD throughout training and peaks at epoch 100 ($\eta = 0.4$), greatly outperforming the best SGD model (+9.65 pp). Training on data with 40% of label noise, evaluating on clean test set, mean of 3 runs, $\alpha = 0.998$.

Table 2: Selection of best-performing methods on CIFAR-10N (Worse) and CIFAR-100N, with 40% label noise in train data, using a ResNet-34. Ours is **highlighted**, all other results are from Wei et al. (2022). Leaderboard available at `http://www.yliuu.com/web-cifarN/Leaderboard.html`

the state-of-the-art. We believe this can be particularly relevant when the presence or the level of label noise is unknown. While specialized (costly) methods need to be justified by heavy label noise, (lightweight) EMA can simply be adopted by default.

### 4.4 Prediction consistency

Training deep neural networks includes multiple sources of randomness, such as batch ordering, initialization and data augmentations. As a result, two independent runs (with exact same algorithm, architecture, training data and hyperparamter configuration) can converge to very different solutions. Even if their accuracy is usually similar, the resulting models will differ in many predictions of individual samples (Jiang et al., 2021b; Bhojanapalli et al., 2021). This prediction disagreement, also known as *churn*, poses a challenge for reproducibility and repeatability in deep learning. Moreover, in real-world systems where the production model is often replaced, it is desirable that the new model, expected to be ever so slightly more accurate, makes predictions consistent with previous models – that is, has a low predictive churn (Jiang et al., 2021a).

We denote the churn between two functions $f_{\theta_1}$ and $f_{\theta_2}$ as the fraction of test samples with different prediction, the lower the better. That is, $\frac{1}{N} \sum_{n=1}^{N} 1[f_{\theta_1}(x_n) \neq f_{\theta_2}(x_n)]$ for the $N$ samples in the test set, where $f(x_n)$ is the top-class predicted. We also propose to use the Jensen-Shannon (JS) divergence as a metric for prediction consistency, which considers the difference in the entire class probability vector. The JS divergence is a symmetrized version of the Kullback–Leibler (KL) divergence defined as $JS(\mathbf{p}\|\mathbf{q}) = 1/2 \, KL(\mathbf{p}\|\mathbf{m}) + 1/2 \, KL(\mathbf{q}\|\mathbf{m})$, where $\mathbf{m} = 1/2(\mathbf{p} + \mathbf{q})$.

A few attempts have been made to reduce churn with algorithmic variations (Bhojanapalli et al., 2021; Jiang et al., 2021a; Madhyastha & Jain, 2019). In our experiments, we train 3 models on independent runs with different seeds and measure the pair-wise churn and JS divergence between their predictions. In Table 3 we compare the prediction consistency of the SGD baseline with the EMA model of lowest validation loss (BN stats recomputed). Using EMA brings a great improvement in consistency between predictions, even outperforming the state of the art (Bhojanapalli et al., 2021), a specialized method that uses co-distillation and has a ×2 training cost. The EMA model consistently reduces the classification churn across different datasets and architectures. Note that we gain a large factor in prediction agreement, as most samples are already correctly predicted by both models, and so don't move. We also find a consistent improvement in the continuous metric of JS divergence.

| | | Test Acc. | | Churn | | JS divergence | |
|---|---|---|---|---|---|---|---|
| | | Baseline | Method | Baseline | Method | Baseline | Method |
| *Method*: EMA (lowest loss) | | | | | | | |
| CIFAR-100 | ResNet-18 | 77.63 | **78.07** | 18.84 ±0.28 | **11.69** ±0.3 | 0.32 ±0.01 | **0.09** ±0.01 |
| | WRN-2810 | 81.07 | **81.90** | 15.69 ±0.09 | **8.88** ±0.04 | 0.10 ±0.0 | **0.055** ±0.0 |
| | VGG-16 | **72.82** | 72.08 | 23.7 ±0.2 | **20.07** ±0.21 | 0.67 ±0.02 | **0.13** ±0.01 |
| CIFAR-10 | ResNet-18 | 95.25 | **95.46** | 3.78 ±0.19 | **2.71** ±0.09 | 0.017 ±0.0 | **0.013** ±0.0 |
| Tiny-ImageNet | ResNet-18 | 66.03 | **67.05** | 29.36 ±0.19 | **15.32** ±0.13 | 0.85 ±0.0 | **0.14** ±0.0 |
| *Method*: Co-distillation KL (Bhojanapalli et al., 2021) | | | | | | | |
| CIFAR-100 | ResNet-56 | 73.26 | **76.53** | 26.77 ±0.26 | **17.09** ±0.3 | - | - |
| CIFAR-10 | ResNet-56 | 93.97 | **94.63** | 5.72 ±0.18 | **4.21** ±0.15 | - | - |

Table 3: Prediction consistency results, measured by Churn and JS divergence, the lower the better. Using an EMA model substantially improves the consistency of predictions between independent runs, achieving a lower churn than methods designed specifically for this goal.

## 4.5 Transfer Learning

In order to assess the quality and generalizability of the learned representations, we test their ability to transfer to other datasets. We investigate whether the implicit regularization effect from stochastic noise promotes the learning of more general representations that generalize across datasets, instead of relying on patterns specific to the training distribution.

We evaluate transfer learning via linear evaluation, similarly to Chen et al. (2020). We use a frozen pretrained model as a feature extractor, all layers but for the last one, and add a linear classification head for another dataset on top. Then, we train the classification head for 50 epochs with SGD with Nesterov momentum of 0.9, without weight decay and with a tuned learning rate of 0.01 without warmup. We do not use EMA on the classification head.

We find that the EMA models learn more general representations which better transfer to other datasets, compared to the SGD baseline. Table 4 shows the results for linear evaluation on the frozen feature extractors, demonstrating that EMA models' representations are more linearly separable when transferred to other tasks. For example, an EMA model pretrained on TinyImagenet achieves a linear evaluation accuracy of 57.78% on CIFAR-100, while the SGD baseline, the same model without EMA, only achieves 52.77%. This result shows that simply adding weight averaging to SGD readily improves the transferability of the features learned. Interestingly, EMA with early stopping at the epoch of lowest validation loss often outperforms the epoch of best accuracy, likely because of early stopping, which is also known as an effective form of implicit regularization.

As expected, since we only train a linear layer, the accuracy of linear evaluation is far from the supervised performance when training an entire model from scratch. Nonetheless, we believe that our results are insightful for understanding the differences between averaged solutions and SGD solutions decaying learning rate to zero, showcasing how averaging promotes the learning of more general representations.

## 4.6 Calibration

Calibration of a model is the property that the predicted probabilities reflect the true likelihood of the ground-truth. While calibrated models are important for high-stakes decision making, for example in medical domains, modern deep neural networks are generally not well-calibrated. Multiple methods have been proposed to improve calibration. Guo et al. (2017) use a post-hoc temperature scaling tuned on a hold-out set. Deep ensembles are also known to improve uncertainty estimation and calibration (Lakshminarayanan et al., 2017; Jiang et al., 2021b), but are an expensive solution. Another class of methods directly optimize for low calibration error during training with auxiliary objectives (Karandikar et al., 2021).

| Pretraining on:
Linear eval on: | Tiny-ImageNet | CIFAR-100
CIFAR-10 | Tiny-ImageNet | CIFAR-10
CIFAR-100 |
|---|---|---|---|---|
| Baseline | $74.61 \pm 0.26$ | $82.10 \pm 0.21$ | $52.77 \pm 0.15$ | $33.72 \pm 1.67$ |
| EMA (best accuracy) | $78.70 \pm 0.45$ | $84.03 \pm 1.07$ | $57.30 \pm 0.39$ | $\mathbf{37.09} \pm 1.06$ |
| EMA (lowest loss) | $\mathbf{79.13} \pm 0.76$ | $\mathbf{85.02} \pm 0.03$ | $\mathbf{57.78} \pm 0.09$ | $36.09 \pm 0.86$ |
| Supervised | | $95.25 \pm 0.11$ | | $77.63 \pm 0.14$ |

Table 4: Linear evaluation on CIFAR-10/100 with a frozen ResNet-18 backbone pretrained on another dataset. Mean and std deviation for 3 seeds. The significant improvements in accuracy using EMA pretrained models indicates that the representations learned are more general and transferable.

In Table 5 we report the test accuracy and calibration error for a baseline model, trained on SGD with Nesterov momentum, and its EMA. We use the Expected Calibration Error (ECE) metric, widely used in the literature. We fix the number of bins to $M = 100$ and compute ECE with equal-mass binning (Nixon et al., 2019). We also report ECE after temperature scaling (TS) as proposed by (Guo et al., 2017). We train on an 80% split of the full training dataset, tune the temperature in the remaining 20% hold-out set, and evaluate on test. For the EMA we use early stopping at the epoch of lowest loss and recompute BN stats after training.

We find that using an EMA considerably reduces the calibration error across all models and datasets tried, compared to the SGD baseline. The improvement that EMA brings seems to be orthogonal to the popular post-hoc operation of temperature scaling, which corrects for an average over/under-confidence. Combining temperature scaling and EMA generally yields the best calibration. We hypothesize that a temporal ensemble of model weights represents a high diversity of solutions, which leads to an improved uncertainty estimation.

| | | Baseline | EMA |
|---|---|---|---|
| ResNet-18
CIFAR-100 | Accuracy | $75.83 \pm 0.05$ | $76.31 \pm 0.28$ |
| | ECE | $11.75 \pm 0.76$ | $9.46 \pm 0.26$ |
| | ECE w/ TS | $4.67 \pm 0.65$ | $\mathbf{3.13} \pm 0.15$ |
| VGG-16
CIFAR-100 | Accuracy | $70.57 \pm 0.11$ | $70.46 \pm 0.18$ |
| | ECE | $20.57 \pm 0.12$ | $8.12 \pm 2.1$ |
| | ECE with TS | $12.17 \pm 0.07$ | $\mathbf{3.64} \pm 0.55$ |
| WideResNet-28-10
CIFAR-100 | Accuracy | $79.29 \pm 0.07$ | $80.12 \pm 0.28$ |
| | ECE | $6.38 \pm 0.25$ | $6.20 \pm 0.53$ |
| | ECE w/ TS | $5.60 \pm 0.1$ | $\mathbf{3.37} \pm 0.23$ |
| ResNet-18
CIFAR-10 | Accuracy | $94.54 \pm 0.22$ | $95.01 \pm 0.08$ |
| | ECE | $3.58 \pm 0.22$ | $1.99 \pm 0.18$ |
| | ECE w/ TS | $1.99 \pm 0.18$ | $\mathbf{1.04} \pm 0.14$ |
| ResNet-18
Tiny-ImageNet | Accuracy | $63.74 \pm 0.12$ | $65.81 \pm 0.2$ |
| | ECE | $12.62 \pm 0.2$ | $8.88 \pm 0.27$ |
| | ECE w/ TS | $\mathbf{3.35} \pm 0.11$ | $3.57 \pm 0.29$ |

Table 5: Expected Calibration Error (ECE) and ECE after Temperature Scaling (TS) results, lower is better. EMA consistently provides better calibrated predictions than the SGD baseline.

## 5    Conclusion

In this work, we have performed a thorough study of weight averaging in deep learning through EMA models, that was lacking in the literature despite its extensive use. We set the goal to answer "What are the properties of weight averaging when training deep neural networks?". While providing a comprehensive understanding of weight averaging in non-convex objectives is a difficult task, we make a first step and gather multiple insights stemming from a rigorous empirical study. We split our contributions in two categories: exploration of training dynamics (Section 3) and properties of the final EMA model (Section 4).

Regarding training dynamics, we first propose a framework to limit the overhead induced by keeping multiple EMA models and tuning the decay rate in one-shot (Section 3.1). We show that averaging with EMA reduces the noise of SGD iterates and allows to maintain high learning rates. In turn, averaging iterates with high stochastic noise leads to a form of implicit regularization that favors learning more general representations (Section 3.2). At the same time, it also allows to spare training epochs by trading noise reduction by learning rate annealing with averaging. Finally, we highlight the striking early performance of EMA models, partly explaining their success as teachers (Section 3.3), and we show that too large averaging windows cannot be used for EMA teachers, since they require recomputing Batch Norm statistics (Section 3.4).

Regarding the final EMA models, we show how they differ from SGD last-iterate solutions and bring an array of benefits. Not only do EMA models generalize better than SGD, on a par with other weight averaging literature methods such as SWA (Section 4.2), but weight averaging also brings robustness to label noise, beating many specialized methods with a much less complex algorithm (Section 4.3). EMA models also improve consistency of predictions across different training runs (Section 4.4), produce more general and transferable representations (Section 4.5), and are better calibrated (Section 4.6).

Admittedly, one limitation of this empirical study is its sole focus on image classification benchmarks. In this work we chose to focus on image classification and explore a wide range of properties of EMA models for this task. The task of image classification has for a long time been a cornerstone for developments in deep learning research, offering mature and trusted benchmarks to compare methods. Nonetheless, it does not guarantee that the properties of EMA models hold for other tasks, which is yet to be explored and remains as future work.

In conclusion, we postulate EMA of weights as an extremely simple yet effective plug-in to improve performance of deep learning models in multiple fronts. We believe this empirical study has immediate practical value, providing a solid case and guidelines for practitioners to add EMA on top of their existing pipelines, while it also sheds some light on the training dynamics of EMA models, which despite their extended use was not covered in the literature.

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

# A   Additional examples of EMA training dynamics

In this section, we provide additional examples of EMA training dynamics, always compared to the momentum SGD baseline, as described in Sec. 4.1. We first discuss the EMA dynamics when the learning rate is decayed in steps, instead of continuously. Then, we provide results on other datasets and network architectures with cosine annealing of the learning rate.

## A.1   EMA dynamics with a step decay

In Sec. 3.2 we discussed the EMA training dynamics when using cosine annealing for the decay of the learning rate. The benefit of a continuous decay is that, since the learning rate controls the strength of implicit regularization, we can tune the level of regularization in the EMA model in one-shot with early stopping. In practice, a very common learning rate schedule is to use a step decay. In Fig. 3 we plot the test accuracy during training when reducing the learning rate by a factor of 5 at epochs $[60, 120, 160]$. In this case the EMA model does *not* outperform the SGD baseline, it only matches its performance towards the end of training, when the learning rate is small enough so that the two models become effectively the same. This is due to a suboptimal choice of the strength of implicit regularization, which we propose to solve with the one-shot tuning by combining cosine decay and early stopping.

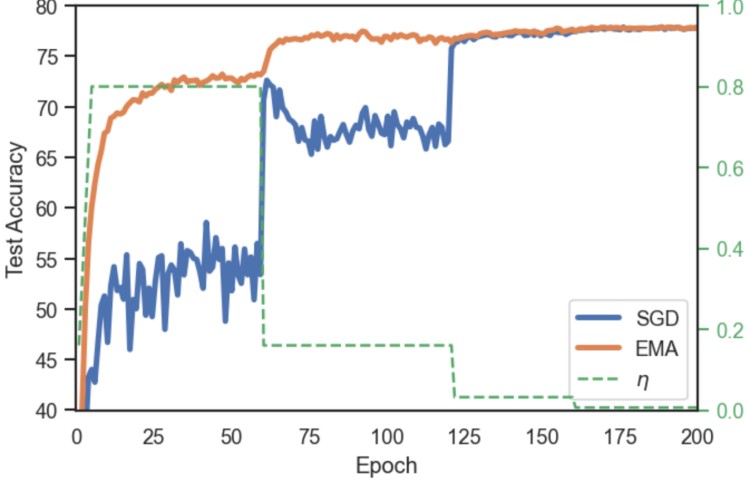

Figure 3: CIFAR-100 on ResNet-18, with step decay of the learning rate by a factor of 5 at epochs $[60, 120, 160]$. At each epoch we report the best EMA out of the 5 parallel EMAs kept, and do not recompute BN stats.

## A.2   EMA dynamics with cosine decay

In this section we include the remaining plots of test accuracy during training for the datasets and architectures that we have based our experiments on. While the evolution of test accuracy during training is different for each case, in all of them we see the same general pattern: the EMA model peaks well before the end of training, when averaging at a higher learning rate $\eta$. As $\eta$ is decreased too much, the effect of implicit regularization is reduced in the EMA and it deteriorates.

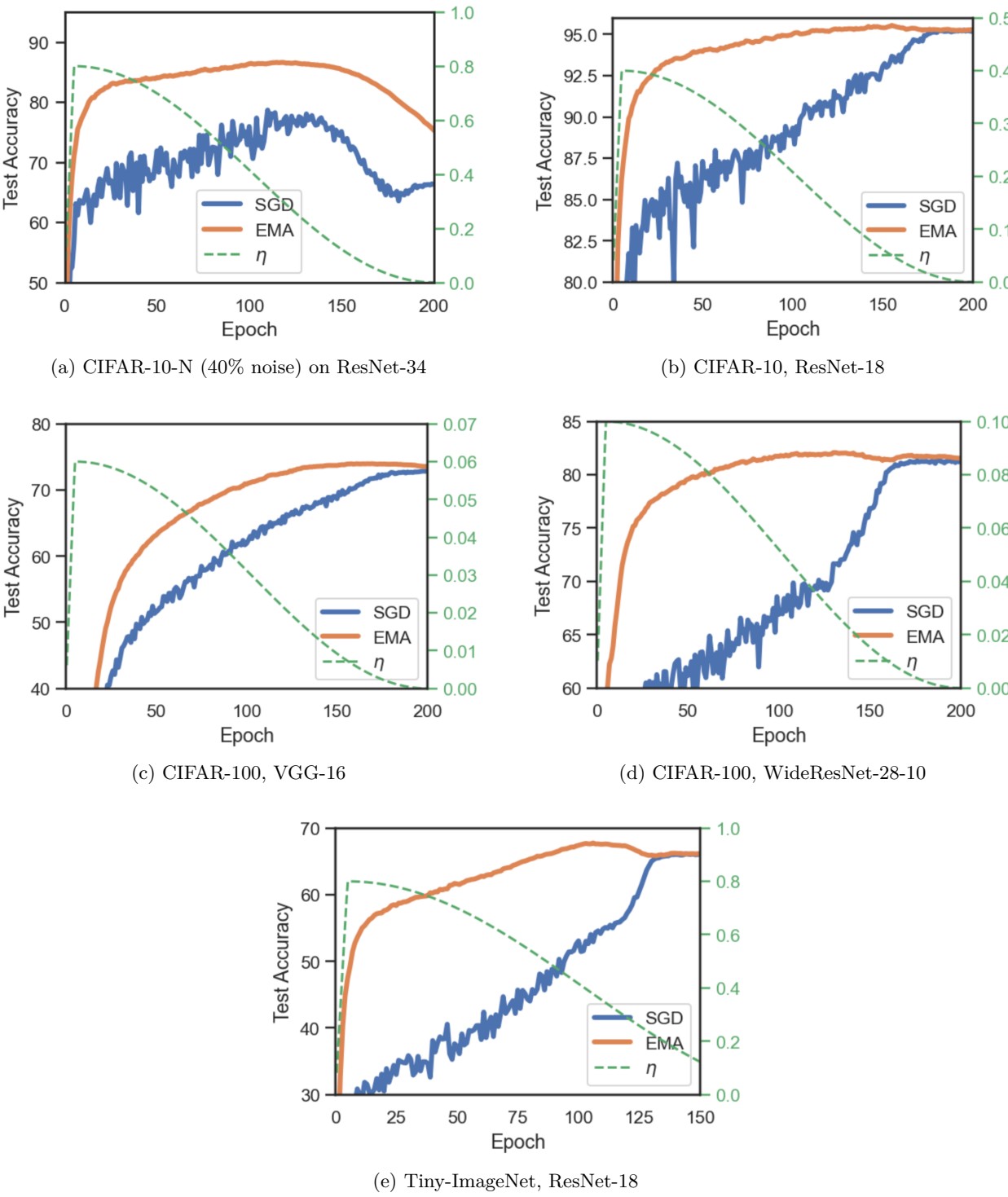

(a) CIFAR-10-N (40% noise) on ResNet-34

(b) CIFAR-10, ResNet-18

(c) CIFAR-100, VGG-16

(d) CIFAR-100, WideResNet-28-10

(e) Tiny-ImageNet, ResNet-18

Figure 4: EMA and momentum SGD training dynamics for the different datasets and models used in our work. Learning rate $(\mu)$ follows a cosine annealing. Training on the full dataset, results are mean of 3 runs. At each epoch we report the best EMA out of the 5 parallel EMAs kept and do not recompute BN stats.

## B  Additional results

In this section, we provide additional results as well as the standard deviation for all results reported in the main text. We also include the results when training on an 80% split of train data, to perform hyperparameter tuning (including early stopping epoch) on the remaining 20% hold-out split. In Table 6 we have a summary of test accuracy and loss in the 80% of training data. The table also includes the average early stopping epoch for EMA among the 3 seeds. Table 6 complements Table 1, which is the subsequent training with 100% of data.

| Architecture | Dataset | Baseline | | | EMA (best acc.) | | | EMA (lowest loss) | | |
|---|---|---|---|---|---|---|---|---|---|---|
| | | Acc. | Loss | Epoch | Acc. | Loss | Epoch | Acc. | Loss | Epoch |
| ResNet-18 | C-100 | 75.83 | 1.09 | 198 | **76.75** | 0.96 | 146 | 76.31 | 0.90 | 124.6 |
| VGG-16 | C-100 | 70.57 | 1.96 | 191.6 | **71.61** | 1.30 | 149.6 | 70.46 | 1.18 | 118 |
| WRN-28-10 | C-100 | 79.29 | 0.86 | 182 | **80.69** | 0.76 | 126.6 | 80.12 | 0.72 | 82 |
| ResNet-18 | C-10 | 94.54 | 0.24 | 189 | **95.06** | 0.19 | 149.3 | 95.01 | 0.17 | 125 |
| ResNet-18 | TinyIN | 63.74 | 1.68 | 148.3 | **66.23** | 1.49 | 101 | 65.81 | 1.45 | 91.3 |

Table 6: Summary of results training on 80% of data. Including epochs to highlight the early stopping and wasteful training of SGD

In the remaining section provide the full results (for both training on 80% and 100% splits) for all the experiments run, namely:

- ResNet-18 on CIFAR-100

- WideResNet-18 on CIFAR-100

- VGG-16 on CIFAR-100

- ResNet-18 on CIFAR-10

- ResNet-18 on Tiny-ImageNet

- ResNet-34 on CIFAR-100-N (label noise)

- ResNet-34 on CIFAR-10-N (label noise)

We report 5 models: the base momentum SGD sequence, its EMA early stopping at best accuracy and lowest loss, as well as the same models after recomputing BN statistics. Other than the mean and standard deviation of the results over 3 independent runs, in Tables 7-20 we also include the best early stopping epochs, EMA decays and learning rates for each of the runs.

| | SGD | EMA (acc.) | EMA (loss) | EMA (acc.) (BN) | EMA (loss) (BN) |
|---|---|---|---|---|---|
| Val Acc. | $75.83 \pm 0.05$ | $76.14 \pm 0.21$ | $75.95 \pm 0.29$ | $76.75 \pm 0.31$ | $76.31 \pm 0.28$ |
| Val Loss | $1.09 \pm 0.04$ | $0.99 \pm 0.04$ | $0.89 \pm 0.0$ | $0.96 \pm 0.03$ | $0.9 \pm 0.01$ |
| Pred Disagr. | $19.49 \pm 0.25$ | $16.95 \pm 0.65$ | $14.82 \pm 0.07$ | $14.04 \pm 0.27$ | $12.78 \pm 0.28$ |
| Pred JS div | $0.299 \pm 0.007$ | $0.243 \pm 0.019$ | $0.14 \pm 0.007$ | $0.157 \pm 0.015$ | $0.107 \pm 0.01$ |
| ECE | $11.75 \pm 0.76$ | $10.96 \pm 0.84$ | $8.39 \pm 0.09$ | $11.0 \pm 0.37$ | $9.46 \pm 0.26$ |
| ECE with TS | $4.67 \pm 0.65$ | $4.17 \pm 0.74$ | $2.72 \pm 0.13$ | $4.39 \pm 0.49$ | $3.13 \pm 0.15$ |
| epochs | $[200, 197, 199]$ | $[143, 139, 156]$ | $[133, 112, 129]$ | $[143, 139, 156]$ | $[133, 112, 129]$ |
| EMA decay | $0$ | $[0.984, 0.984, 0.968]$ | $[0.984, 0.984, 0.984]$ | $0.998$ | $0.998$ |
| LR | $[1.2, 0.8, 1.2]$ | $[1.2, 0.8, 1.2]$ | $[1.2, 0.8, 1.2]$ | $[1.2, 0.8, 1.2]$ | $[1.2, 0.8, 1.2]$ |

Table 7: ResNet-18 on CIFAR-100 results for 3 runs. Training on 80% split, evaluation on hold-out 20% split. (BN) denotes recomputation of Batch Norm statistics

| | SGD | EMA (acc.) | EMA (loss) | EMA (acc.) (BN) | EMA (loss) (BN) |
|---|---|---|---|---|---|
| Test Accuracy | $77.63 \pm 0.14$ | $77.99 \pm 0.04$ | $77.69 \pm 0.28$ | $78.54 \pm 0.28$ | $78.07 \pm 0.29$ |
| Test Loss | $1.02 \pm 0.03$ | $0.84 \pm 0.03$ | $0.81 \pm 0.01$ | $0.84 \pm 0.02$ | $0.82 \pm 0.0$ |
| Pred Disagr. | $18.84 \pm 0.28$ | $14.27 \pm 0.08$ | $13.24 \pm 0.05$ | $12.22 \pm 0.24$ | $11.69 \pm 0.3$ |
| Pred JS div | $0.325 \pm 0.007$ | $0.156 \pm 0.01$ | $0.113 \pm 0.007$ | $0.124 \pm 0.012$ | $0.087 \pm 0.003$ |
| ECE | $11.47 \pm 0.76$ | $8.51 \pm 1.04$ | $6.86 \pm 0.69$ | $8.99 \pm 0.68$ | $7.96 \pm 0.48$ |
| ECE with TS | $7.16 \pm 1.09$ | $6.39 \pm 0.69$ | $5.2 \pm 0.63$ | $6.98 \pm 0.64$ | $5.77 \pm 0.66$ |
| epochs | [200, 200, 200] | [143, 139, 155] | [133, 112, 147] | [143, 139, 155] | [133, 112, 147] |
| EMA decay | 0 | [0.992, 0.992, 0.992] | [0.992, 0.984, 0.992] | 0.998 | 0.998 |
| LR | [1.2, 0.8, 1.2] | [1.2, 0.8, 1.2] | [1.2, 0.8, 1.2] | [1.2, 0.8, 1.2] | [1.2, 0.8, 1.2] |

Table 8: ResNet-18 on CIFAR-100 results for 3 runs. Training on full training set. (BN) denotes recomputation of Batch Norm statistics

| | SGD | EMA (acc.) | EMA (loss) | EMA (acc.) (BN) | EMA (loss) (BN) |
|---|---|---|---|---|---|
| Val Accuracy | $79.29 \pm 0.07$ | $79.91 \pm 0.21$ | $79.19 \pm 0.56$ | $80.69 \pm 0.23$ | $80.12 \pm 0.28$ |
| Val Loss | $0.86 \pm 0.01$ | $0.82 \pm 0.01$ | $0.78 \pm 0.01$ | $0.76 \pm 0.01$ | $0.72 \pm 0.01$ |
| Pred Disagr. | $17.31 \pm 0.24$ | $13.44 \pm 0.24$ | $11.92 \pm 0.21$ | $11.43 \pm 0.11$ | $10.88 \pm 0.04$ |
| Pred JS div | $0.108 \pm 0.002$ | $0.135 \pm 0.011$ | $0.094 \pm 0.006$ | $0.095 \pm 0.003$ | $0.069 \pm 0.003$ |
| ECE | $6.38 \pm 0.25$ | $8.66 \pm 0.07$ | $6.32 \pm 0.73$ | $8.44 \pm 0.06$ | $6.2 \pm 0.53$ |
| ECE with TS | $5.6 \pm 0.1$ | $3.26 \pm 0.22$ | $3.29 \pm 0.03$ | $2.99 \pm 0.05$ | $3.37 \pm 0.23$ |
| epochs | [175, 175, 196] | [126, 131, 123] | [70, 84, 92] | [126, 131, 123] | [70, 84, 92] |
| EMA decay | 0 | [0.992, 0.984, 0.984] | [0.992, 0.984, 0.984] | 0.998 | 0.998 |
| LR | [0.1, 0.1, 0.1] | [0.1, 0.1, 0.1] | | | |

Table 9: WideResNet-28-10 on CIFAR-100 results for 3 runs. Training on 80% split, evaluation on hold-out 20% split. (BN) denotes recomputation of Batch Norm statistics

| | SGD | EMA (acc.) | EMA (loss) | EMA (acc.) (BN) | EMA (loss) (BN) |
|---|---|---|---|---|---|
| Test Accuracy | $81.07 \pm 0.12$ | $81.88 \pm 0.09$ | $81.09 \pm 0.32$ | $82.73 \pm 0.16$ | $81.91 \pm 0.33$ |
| Test Loss | $0.78 \pm 0.01$ | $0.72 \pm 0.01$ | $0.69 \pm 0.0$ | $0.67 \pm 0.01$ | $0.64 \pm 0.0$ |
| Pred Disagr. | $15.69 \pm 0.09$ | $11.62 \pm 0.2$ | $10.35 \pm 0.41$ | $9.95 \pm 0.21$ | $8.88 \pm 0.04$ |
| Pred JS div | $0.1 \pm 0.002$ | $0.117 \pm 0.005$ | $0.078 \pm 0.009$ | $0.079 \pm 0.002$ | $0.055 \pm 0.002$ |
| ECE | $4.88 \pm 0.1$ | $8.06 \pm 0.29$ | $6.52 \pm 0.25$ | $6.78 \pm 0.29$ | $5.03 \pm 0.29$ |
| ECE with TS | $8.24 \pm 0.5$ | $4.33 \pm 0.18$ | $4.05 \pm 0.13$ | $3.37 \pm 0.31$ | $2.91 \pm 0.04$ |
| epochs | [200, 200, 200] | [126, 131, 123] | [70, 84, 92] | [126, 131, 123] | [70, 84, 92] |
| EMA decay | 0 | [0.984, 0.992, 0.992] | [0.984, 0.992, 0.984] | 0.998 | 0.998 |
| LR | [0.1, 0.1, 0.1] | [0.1, 0.1, 0.1] | [0.1, 0.1, 0.1] | | |

Table 10: WideResNet-28-10 on CIFAR-100 results for 3 runs. Training on full training set. (BN) denotes recomputation of Batch Norm statistics

| | SGD | EMA (acc.) | EMA (loss) |
|---|---|---|---|
| Val Accuracy | $70.57 \pm 0.11$ | $71.61 \pm 0.25$ | $70.46 \pm 0.18$ |
| Val Loss | $1.96 \pm 0.02$ | $1.3 \pm 0.03$ | $1.18 \pm 0.03$ |
| Pred Disagr. | $26.0 \pm 0.36$ | $22.76 \pm 0.07$ | $23.21 \pm 0.66$ |
| Pred JS div | $0.829 \pm 0.023$ | $0.322 \pm 0.021$ | $0.233 \pm 0.024$ |
| ECE | $20.57 \pm 0.12$ | $14.49 \pm 0.65$ | $8.12 \pm 2.1$ |
| ECE with TS | $12.17 \pm 0.07$ | $4.63 \pm 0.44$ | $3.64 \pm 0.55$ |
| epochs | [199, 186, 190] | [150, 146, 153] | [113, 121, 120] |
| EMA decay | 0 | [0.998, 0.998, 0.998] | [0.996, 0.984, 0.998] |
| LR | [0.05, 0.05, 0.05] | [0.05, 0.05, 0.05] | [0.05, 0.05, 0.05] |

Table 11: VGG-16 on CIFAR-100 results for 3 runs. Training on 80% split, evaluation on hold-out 20% split. (BN) denotes recomputation of Batch Norm statistics

| | SGD | EMA (acc.) | EMA (loss) | EMA (acc.) (BN) | EMA (loss) (BN) |
|---|---|---|---|---|---|
| Test Accuracy | $72.82 \pm 0.17$ | $73.64 \pm 0.13$ | $72.3 \pm 0.19$ | $73.62 \pm 0.13$ | $72.08 \pm 0.06$ |
| Test Loss | $1.77 \pm 0.03$ | $1.17 \pm 0.04$ | $1.1 \pm 0.01$ | $1.13 \pm 0.02$ | $1.06 \pm 0.01$ |
| Pred Disagr. | $23.7 \pm 0.2$ | $21.42 \pm 0.15$ | $22.12 \pm 0.17$ | $20.89 \pm 0.25$ | $20.07 \pm 0.21$ |
| Pred JS div | $0.676 \pm 0.023$ | $0.28 \pm 0.016$ | $0.234 \pm 0.01$ | $0.238 \pm 0.006$ | $0.134 \pm 0.005$ |
| ECE | $19.06 \pm 0.24$ | $13.04 \pm 1.09$ | $9.1 \pm 0.4$ | $12.29 \pm 0.56$ | $5.66 \pm 0.74$ |
| ECE with TS | $15.85 \pm 0.2$ | $8.72 \pm 1.04$ | $4.89 \pm 0.26$ | $8.01 \pm 0.51$ | $3.15 \pm 0.25$ |
| epochs | [200, 200, 200] | [149, 146, 149] | [113, 121, 120] | [149, 146, 149] | [113, 121, 120] |
| EMA decay | 0 | [0.996, 0.998, 0.996] | [0.984, 0.992, 0.984] | 0.998 | 0.998 |

Table 12: VGG-16 on CIFAR-100 results for 3 runs. Training on full training set. The only difference with (BN) in this case, since VGG-16 does not have BN, is the use of the larger decay

| | SGD | EMA (acc.) | EMA (loss) | EMA (acc.) (BN) | EMA (loss) (BN) |
|---|---|---|---|---|---|
| Val Accuracy | $94.54 \pm 0.22$ | $94.77 \pm 0.16$ | $94.61 \pm 0.12$ | $95.06 \pm 0.15$ | $95.01 \pm 0.08$ |
| Val Loss | $0.24 \pm 0.01$ | $0.21 \pm 0.02$ | $0.19 \pm 0.01$ | $0.19 \pm 0.02$ | $0.17 \pm 0.0$ |
| Pred Disagr. | $4.5 \pm 0.13$ | $4.12 \pm 0.32$ | $3.57 \pm 0.15$ | $3.5 \pm 0.29$ | $2.99 \pm 0.06$ |
| Pred JS div | $0.017 \pm 0.0$ | $0.054 \pm 0.028$ | $0.018 \pm 0.003$ | $0.04 \pm 0.018$ | $0.013 \pm 0.001$ |
| ECE | $3.58 \pm 0.22$ | $3.01 \pm 0.49$ | $2.47 \pm 0.24$ | $2.65 \pm 0.56$ | $1.99 \pm 0.18$ |
| ECE with TS | $1.99 \pm 0.18$ | $1.58 \pm 0.35$ | $1.18 \pm 0.13$ | $1.44 \pm 0.29$ | $1.04 \pm 0.14$ |
| epochs | [187, 184, 196] | [160, 162, 126] | [124, 139, 113] | [160, 162, 126] | [124, 139, 113] |
| EMA decay | 0 | [0.992, 0.984, 0.992] | [0.968, 0.992, 0.992] | 0.998 | 0.998 |
| LR | [0.4, 0.8, 0.4] | [0.4, 0.8, 0.4] | [0.4, 0.8, 0.4] | [0.4, 0.8, 0.4] | [0.4, 0.8, 0.4] |

Table 13: ResNet-18 on CIFAR-10 results for 3 runs. Training on 80% split, evaluation on hold-out 20% split. (BN) denotes recomputation of Batch Norm statistics

| | SGD | EMA (acc.) | EMA (loss) | EMA (acc.) (BN) | EMA (loss) (BN) |
|---|---|---|---|---|---|
| Test Accuracy | $95.25 \pm 0.11$ | $95.39 \pm 0.07$ | $95.24 \pm 0.04$ | $95.62 \pm 0.11$ | $95.46 \pm 0.18$ |
| Test Loss | $0.22 \pm 0.0$ | $0.19 \pm 0.02$ | $0.16 \pm 0.0$ | $0.17 \pm 0.02$ | $0.15 \pm 0.0$ |
| Pred Disagr. | $3.78 \pm 0.19$ | $3.35 \pm 0.15$ | $3.01 \pm 0.18$ | $3.03 \pm 0.14$ | $2.71 \pm 0.09$ |
| Pred JS div | $0.017 \pm 0.0$ | $0.044 \pm 0.021$ | $0.013 \pm 0.0$ | $0.034 \pm 0.016$ | $0.01 \pm 0.0$ |
| ECE | $3.2 \pm 0.09$ | $2.62 \pm 0.4$ | $2.03 \pm 0.16$ | $2.33 \pm 0.43$ | $1.7 \pm 0.06$ |
| ECE with TS | $2.46 \pm 0.08$ | $1.94 \pm 0.35$ | $1.56 \pm 0.1$ | $1.69 \pm 0.38$ | $1.2 \pm 0.09$ |
| epochs | [200, 200, 200] | [160, 162, 126] | [124, 139, 113] | [160, 162, 126] | [124, 139, 113] |
| EMA decay | 0 | [0.992, 0.996, 0.992] | [0.992, 0.992, 0.992] | 0.998 | 0.998 |
| LR | [0.4, 0.8, 0.4] | [0.4, 0.8, 0.4] | [0.4, 0.8, 0.4] | [0.4, 0.8, 0.4] | [0.4, 0.8, 0.4] |

Table 14: ResNet-18 on CIFAR-10 results for 3 runs. Training on full training set. (BN) denotes recomputation of Batch Norm statistics

| | SGD | EMA (acc.) | EMA (loss) | EMA (acc.) (BN) | EMA (loss) (BN) |
|---|---|---|---|---|---|
| Val Accuracy | $63.74 \pm 0.12$ | $65.04 \pm 0.18$ | $65.29 \pm 0.13$ | $66.23 \pm 0.11$ | $65.81 \pm 0.2$ |
| Val Loss | $1.68 \pm 0.01$ | $1.48 \pm 0.02$ | $1.45 \pm 0.0$ | $1.49 \pm 0.01$ | $1.45 \pm 0.01$ |
| Pred Disagr. | $30.66 \pm 0.09$ | $22.94 \pm 0.46$ | $21.65 \pm 0.57$ | $19.3 \pm 0.18$ | $18.16 \pm 0.08$ |
| Pred JS div | $0.785 \pm 0.01$ | $0.354 \pm 0.015$ | $0.269 \pm 0.01$ | $0.276 \pm 0.011$ | $0.199 \pm 0.006$ |
| ECE | $12.62 \pm 0.2$ | $9.22 \pm 0.5$ | $6.23 \pm 0.17$ | $10.68 \pm 0.31$ | $8.88 \pm 0.27$ |
| ECE with TS | $3.35 \pm 0.11$ | $3.21 \pm 0.04$ | $4.68 \pm 0.15$ | $3.26 \pm 0.1$ | $3.57 \pm 0.29$ |
| epochs | [147, 150, 148] | [103, 99, 101] | [92, 90, 92] | [103, 99, 101] | [92, 90, 92] |
| EMA decay | 0 | [0.992, 0.992, 0.992] | [0.992, 0.984, 0.984] | 0.998 | 0.998 |

Table 15: ResNet-18 on Tiny-ImageNet results for 3 runs. Training on 80% split, evaluation on hold-out 20% split. (BN) denotes recomputation of Batch Norm statistics

| | SGD | EMA (acc.) | EMA (loss) | EMA (acc.) (BN) | EMA (loss) (BN) |
|---|---|---|---|---|---|
| Test Accuracy | $66.03 \pm 0.26$ | $67.56 \pm 0.16$ | $66.51 \pm 0.28$ | $67.97 \pm 0.14$ | $67.06 \pm 0.18$ |
| Test Loss | $1.6 \pm 0.01$ | $1.34 \pm 0.0$ | $1.36 \pm 0.01$ | $1.35 \pm 0.01$ | $1.36 \pm 0.0$ |
| Pred Disagr. | $29.36 \pm 0.19$ | $19.84 \pm 0.28$ | $19.89 \pm 0.04$ | $16.67 \pm 0.24$ | $15.35 \pm 0.13$ |
| Pred JS div | $0.85 \pm 0.003$ | $0.261 \pm 0.009$ | $0.23 \pm 0.012$ | $0.192 \pm 0.005$ | $0.14 \pm 0.001$ |
| ECE | $13.09 \pm 0.07$ | $6.47 \pm 0.39$ | $4.94 \pm 0.19$ | $8.31 \pm 0.32$ | $7.14 \pm 0.33$ |
| ECE with TS | $5.97 \pm 0.05$ | $5.4 \pm 0.28$ | $3.93 \pm 0.2$ | $5.43 \pm 0.39$ | $4.07 \pm 0.26$ |
| epochs | [150, 150, 150] | [103, 99, 101] | [92, 89, 92] | [103, 99, 101] | [92, 89, 92] |
| EMA decay | 0 | [0.992, 0.992, 0.992] | [0.968, 0.984, 0.984] | 0.998 | 0.998 |

Table 16: ResNet-18 on Tiny-ImageNet results for 3 runs. Training on full training set. (BN) denotes recomputation of Batch Norm statistics

| | SGD | EMA (acc.) | EMA (loss) | EMA (acc.) (BN) | EMA (loss) (BN) |
|---|---|---|---|---|---|
| Val Accuracy | $54.37 \pm 0.18$ | $61.95 \pm 0.07$ | $61.89 \pm 0.43$ | $63.09 \pm 0.13$ | $62.76 \pm 0.28$ |
| Val Loss | $2.43 \pm 0.02$ | $1.37 \pm 0.02$ | $1.37 \pm 0.01$ | $1.32 \pm 0.01$ | $1.32 \pm 0.01$ |
| Pred Disagr. | $38.81 \pm 0.1$ | $23.04 \pm 0.56$ | $20.0 \pm 0.09$ | $19.04 \pm 0.36$ | $17.26 \pm 0.23$ |
| Pred JS div | $0.606 \pm 0.006$ | $0.082 \pm 0.006$ | $0.048 \pm 0.002$ | $0.061 \pm 0.002$ | $0.044 \pm 0.001$ |
| ECE | $23.89 \pm 0.05$ | $2.9 \pm 0.28$ | $4.03 \pm 0.2$ | $5.07 \pm 0.44$ | $3.53 \pm 0.19$ |
| ECE with TS | $7.29 \pm 0.28$ | $14.41 \pm 0.3$ | $16.43 \pm 0.25$ | $11.1 \pm 0.34$ | $12.91 \pm 0.33$ |
| epochs | [196, 192, 184] | [91, 82, 85] | [60, 63, 51] | [91, 82, 85] | [60, 63, 51] |
| EMA decay | 0 | [0.968, 0.968, 0.984] | [0.984, 0.984, 0.968] | 0.998 | 0.998 |

Table 17: ResNet-34 on CIFAR-100-N (40% noisy labels) results for 3 runs. Training on 80% split, evaluation on hold-out 20% split. (BN) denotes recomputation of Batch Norm statistics

| | SGD | EMA (acc.) | EMA (loss) | EMA (acc.) (BN) | EMA (loss) (BN) |
|---|---|---|---|---|---|
| Test Accuracy | $55.47 \pm 0.35$ | $64.18 \pm 0.18$ | $62.95 \pm 0.34$ | $65.15 \pm 0.2$ | $63.95 \pm 0.12$ |
| Test Loss | $2.43 \pm 0.03$ | $1.28 \pm 0.01$ | $1.33 \pm 0.01$ | $1.23 \pm 0.0$ | $1.27 \pm 0.01$ |
| Pred Disagr. | $38.03 \pm 0.21$ | $18.2 \pm 0.18$ | $18.48 \pm 0.32$ | $15.17 \pm 0.1$ | $14.87 \pm 0.3$ |
| Pred JS div | $0.67 \pm 0.006$ | $0.045 \pm 0.002$ | $0.039 \pm 0.002$ | $0.036 \pm 0.001$ | $0.031 \pm 0.001$ |
| ECE | $24.76 \pm 0.44$ | $4.36 \pm 0.36$ | $5.65 \pm 0.29$ | $3.24 \pm 0.11$ | $3.14 \pm 0.07$ |
| ECE with TS | $17.1 \pm 0.44$ | $11.72 \pm 0.42$ | $13.81 \pm 0.21$ | $9.8 \pm 0.38$ | $11.32 \pm 0.4$ |
| epochs | [200, 200, 200] | [91, 82, 85] | [61, 64, 52] | [91, 82, 85] | [61, 64, 52] |
| EMA decay | 0 | [0.984, 0.984, 0.984] | [0.968, 0.984, 0.968] | 0.998 | 0.998 |

Table 18: ResNet-34 on CIFAR-100-N (40% noisy labels) results for 3 runs. Training on full training set. (BN) denotes recomputation of Batch Norm statistics

| | SGD | EMA (acc.) | EMA (loss) | EMA (acc.) (BN) | EMA (loss) (BN) |
|---|---|---|---|---|---|
| Val Accuracy | $66.22 \pm 0.47$ | $85.09 \pm 0.14$ | $85.04 \pm 0.25$ | $85.49 \pm 0.12$ | $85.57 \pm 0.32$ |
| Val Loss | $2.03 \pm 0.05$ | $0.64 \pm 0.01$ | $0.65 \pm 0.0$ | $0.55 \pm 0.01$ | $0.57 \pm 0.0$ |
| Pred Disagr. | $37.2 \pm 0.41$ | $6.54 \pm 0.2$ | $6.38 \pm 0.09$ | $5.7 \pm 0.46$ | $5.28 \pm 0.25$ |
| Pred JS div | $0.152 \pm 0.002$ | $0.001 \pm 0.0$ | $0.001 \pm 0.0$ | $0.001 \pm 0.0$ | $0.0 \pm 0.0$ |
| ECE | $24.85 \pm 0.53$ | $24.48 \pm 0.38$ | $25.07 \pm 0.09$ | $18.29 \pm 0.32$ | $19.07 \pm 0.14$ |
| ECE with TS | $22.74 \pm 0.51$ | $31.17 \pm 0.32$ | $31.91 \pm 0.01$ | $30.21 \pm 0.36$ | $30.96 \pm 0.13$ |
| epochs | [200, 200, 200] | [124, 109, 117] | [98, 98, 102] | [124, 109, 117] | [98, 98, 102] |
| EMA decay | 0 | [0.996, 0.984, 0.992] | [0.984, 0.984, 0.984] | 0.998 | 0.998 |

Table 19: ResNet-34 on CIFAR-10-N (Worse, 40% noisy labels) results for 3 runs. Training on 80% split, evaluation on hold-out 20% split. (BN) denotes recomputation of Batch Norm statistics

| | SGD | EMA (acc.) | EMA (loss) | EMA (acc.) (BN) | EMA (loss) (BN) |
|---|---|---|---|---|---|
| Test Accuracy | $78.09 \pm 0.23$ | $86.4 \pm 0.13$ | $86.19 \pm 0.12$ | $86.71 \pm 0.17$ | $86.35 \pm 0.09$ |
| Test Loss | $0.82 \pm 0.02$ | $0.64 \pm 0.0$ | $0.66 \pm 0.0$ | $0.56 \pm 0.0$ | $0.57 \pm 0.0$ |
| Pred Disagr. | $23.29 \pm 0.42$ | $7.46 \pm 0.17$ | $7.55 \pm 0.53$ | $6.63 \pm 0.28$ | $5.62 \pm 0.18$ |
| Pred JS div | $0.006 \pm 0.0$ | $0.001 \pm 0.0$ | $0.001 \pm 0.0$ | $0.001 \pm 0.0$ | $0.001 \pm 0.0$ |
| ECE | $20.79 \pm 2.31$ | $21.66 \pm 0.2$ | $23.26 \pm 0.38$ | $15.86 \pm 0.64$ | $17.62 \pm 0.42$ |
| ECE with TS | $30.22 \pm 0.75$ | $32.93 \pm 0.08$ | $33.3 \pm 0.22$ | $30.94 \pm 0.34$ | $31.77 \pm 0.37$ |
| epochs | [111, 118, 113] | [124, 109, 117] | [98, 98, 102] | [124, 109, 117] | [98, 98, 102] |
| EMA decay | 0 | [0.996, 0.984, 0.984] | [0.984, 0.992, 0.968] | 0.998 | 0.998 |

Table 20: ResNet-34 on CIFAR-10-N (Worse, 40% noisy labels) results for 3 runs. Training on full training set. (BN) denotes recomputation of Batch Norm statistics

## C   Bootstrapping on EMA

In Figure 5 we compare the normal EMA usage (*i.e.*, to apply a slow-moving average of the SGD sequence, always outside the training loop) vs. bootstrapping the SGD model (*i.e.*, student) once per epoch with the averaged parameters of the EMA. As the EMA model performs better in the early stages of training, we test whether using it to bootstrap the training parameters expedites training. Nonetheless, we find the opposite effect, bootstrapping not only does not help but it actually decreases performance. In the figure we can see how both the student model and EMA model validation accuracy during training are worse than without bootstrap. The same is true for the final validation accuracy of both the student, as regular momentum SGD achieves 75.83% and the bootstrapped SGD only 74.47%. A tentative explanation for the failure of bootstrapping SGD with its EMA, is that the EMA model is only a good point in the local neighborhood of the SGD sequence (as it reduces the noise), and not an advancement into a better neighborhood of the landscape, which is what ultimately is important to achieve a better final model.

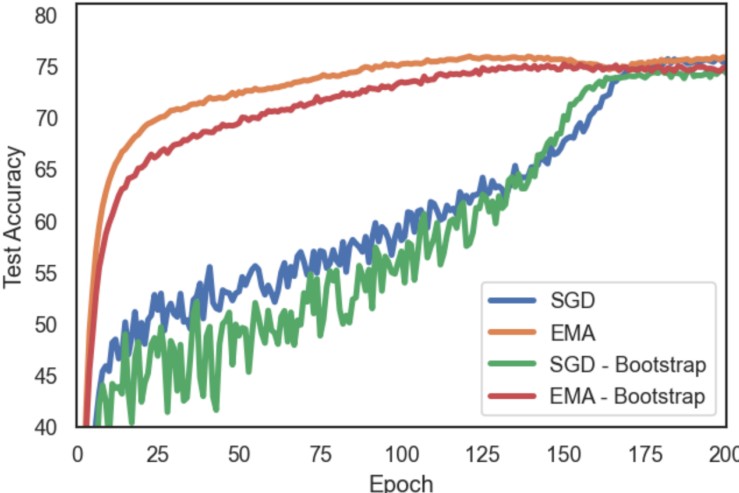

Figure 5: CIFAR-100 on ResNet-18, training on 80% split and evaluation on hold-out 20% split. The EMA is sampled every $T = 16$ steps and has a decay $\alpha = 0.992$. We compare the performance of regular SGD baseline (and its EMA) to a bootstrapped SGD (and its EMA). In particular, we bootstrap the SGD iterate with the EMA weights once every epoch.

# D    Learning rate tuning for SGD vs EMA

We explore the choice of learning rate for EMA and SGD, and investigate if the optimal value is significantly different. Despite the differences in dynamics and final solution between the EMA and the base SGD sequence, we find an alignment when tuning the initial learning rate. During our hyperparameter search we find the same best initial learning rate $\eta$ for both sequences. We hypothesize that the tuning of the learning rate affects mostly the early training stage, which allows for fast progress and benefits generalization, and is not dependent on the technique used to reduce noise for convergence at the end of training (either decaying $\eta$ or averaging). In Fig. 6 we plot the evolution of validation accuracy during training. We observe a trend where the lower learning rate has a better accuracy in the first epochs, but peaks lower than higher learning rates.

| $\eta$ | SGD Acc. | EMA Acc. |
|-----|------|------|
| 0.4 | 75.3 | 75.4 |
| 0.8 | 75.8 | **76.1** |
| 1.2 | **75.9** | **76.1** |
| 1.6 | 75.1 | 75.7 |

(a) ResNet-18

| $\eta$ | SGD Acc. | EMA Acc. |
|------|------|------|
| 0.05 | 78.3 | 79.5 |
| 0.1 | **79.0** | **80.0** |
| 0.2 | **79.0** | 79.8 |
| 0.4 | 76.9 | 77.5 |

(b) WideResNet-28-10

Table 21: CIFAR-100 Validation Accuracy for SGD and EMA for different initial learning rates $\eta$. We train on 80% split of training data and use the remaining 20% hold-out set for evaluation.

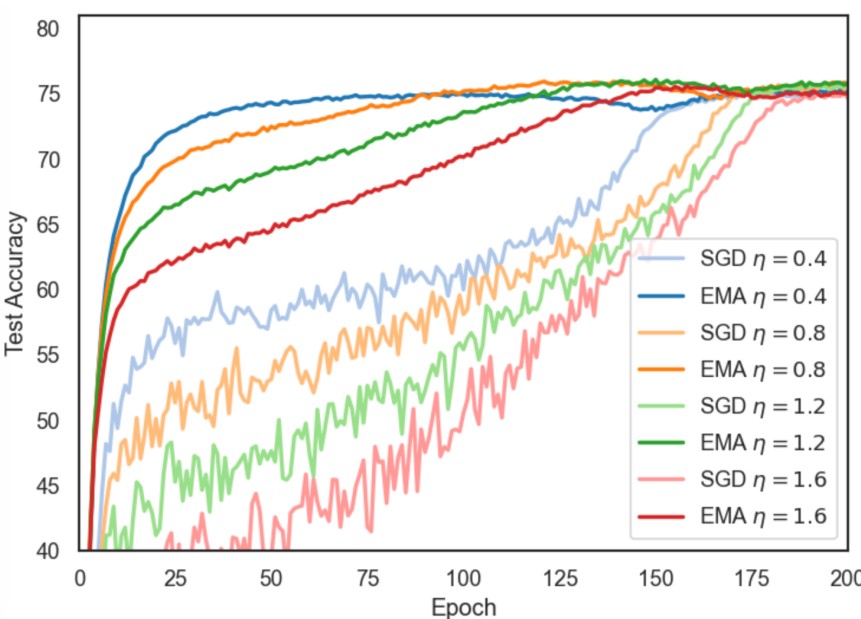

Figure 6: SGD and EMA validation accuracy for different initial values of learning rate $\eta$.

# E   Additional Label Noise Experiments

## E.1   Memorization of noisy labels

We validate this explanation with Fig. 7, where we see that the memorization of noisy labels is lower in the EMA model with respect to train accuracy on the clean labels. For instance, when EMA reaches 90% clean accuracy at epoch 99, it has memorized 28.8% of the noisy labels, while for the SGD model, in epoch 142, the noisy accuracy was of 68.2%. Another interesting observation is that the best EMA performance is reached at epoch 100, which is when its memorization of noise starts to increase rapidly.

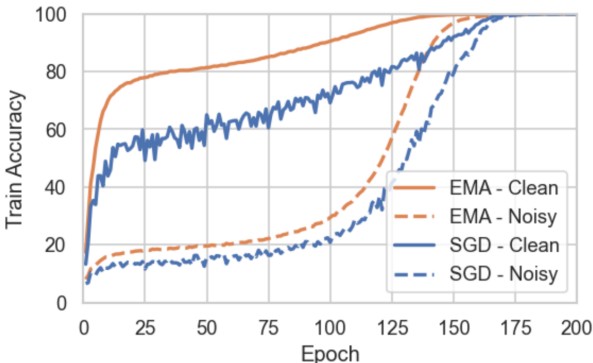

Figure 7: CIFAR-100N on ResNet-34. Accuracy on *training* data during training, split into *Noisy* (40% of data wrong labels) *Clean* (remaining 60%). Both models end up by memorizing all of the noisy labels, but the EMA model fits less noise relative the accuracy on the clean samples.

## E.2   Continued training at constant Learning Rate

In the experiments training with label noise (Sec. 4.3) we found that the effect of implicit regularization in the EMA was very large, preventing memorization of noisy labels and improving test accuracy. However, to make sure that memorization and overfitting occur due to the decay of the learning rate, and not simply because of continued training, we perform the following ablation study. We present the test accuracy during training for a model with learning rate decay vs keeping the learning rate constant after the best epoch.

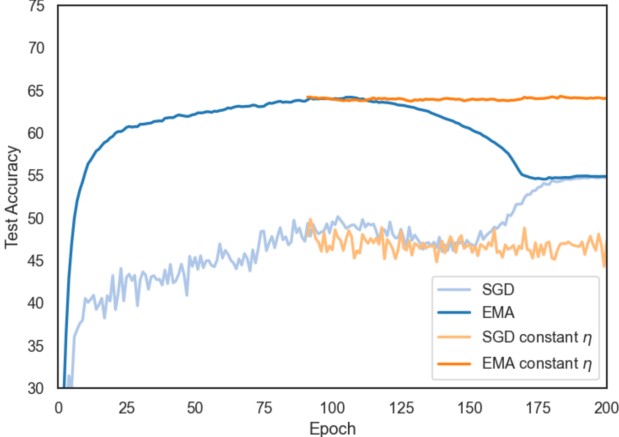

Figure 8: Test Accuracy with constant learning rate after stopping epoch (for EMA (acc.)) vs with cosine decay. Overfitting is due to learning rate decay, not continued training. Experiments with ResNet-18 on Cifar-100 (40% noise) full training set. Sliding window of 5 for smoothing of curves.

## F  Detailed experimental setup

Our experimental set up follows these steps for hyperparameter tuning:

- Split train set into train/validation as 80/20.

- Tune hyperparamters (learning rate, early stopping epochs) on validation set. Note that for EMA we distinguish between two early stopping criteria: best accuracy and lowest loss.

- Finally, train again on 100% on train data using the hyperparameters found on the validation set. Report final performance on the test set, which was not used for hyperparameter tuning. Note that this should be done for all deep learning methods, even if it is often not the case.

We fix the number of training epochs, batch size and weight decay. As for the EMA, we search for the best decay by keeping 5 parallel EMAs with $\tau \in [0.968, 0.984, 0.992, 0.996, 0.998]$. We warmup the EMA decay in the first steps as $\min(\alpha, \frac{t+1}{t+10})$. EMA sampling every of $T = 16$ steps (note that this affects the effective decay, see below). In Table 22 we include a summary of the hyperparamter configuration. The best values for the hyperparmeters tuned on the validation set are reported in App. B. For all experiments, we report the mean of 3 independent runs.

| Setting | Value |
|---|---|
| Optimizer | SGD with Nesterov momentum |
| Momentum | 0.9 |
| Learning rate | Tuned on validation set |
| Early stopping epochs | Tuned on validation set |
| Weight Decay | ResNet: $1 \times 10^{-4}$. WideResNet, VGG-16: $5 \times 10^{-4}$ |
| Batch size | 128 |
| Epochs | CIFAR-10/100: 200. Tiny-ImageNet: 150 |
| EMA decays | $[0.968, 0.984, 0.992, 0.996, 0.998]$ |
| EMA sampling period | $T = 16$ |

Table 22: Summary of hyperparamter configuration

The decay rate $\alpha$ for the exponential moving average governs how fast past iterations are forgotten. For the use of EMA in deep learning, we find empirically that sampling at a period $T > 1$ can reduce overhead without impact on the results. We use $T = 16$ in our implementation. However, it is important to note that changing the sampling period will affect the decay (past iterates will be reweighted once every $T$ steps only). For this reason, to keep the same effective decay rate, the decay of the EMA sequence has to be updated as $\alpha' = \alpha^T$. In Tab. 23 we include a summary of the decay rates we used at $T = 16$ and their equivalent decay rate if sampling at $T = 1$.

| $T = 1$ | $T = 16$ |
|---|---|
| 0.999875 | 0.998 |
| 0.99975 | 0.996 |
| 0.9995 | 0.992 |
| 0.999 | 0.984 |
| 0.998 | 0.968 |

Table 23: Equivalence of EMA decay rate $\alpha$ for different sampling periods.

## G  Sensitivity analysis to EMA decay rate $\alpha$

The decay rate $\alpha$ is a key hyperparameter in EMA models. In this section we include a sensitivity analysis for the range of decay rates explored, which are $\tau \in [0.968, 0.984, 0.992, 0.996, 0.998]$. We chose this range, asymptotically approaching 1, since previous works and early experiments have shown that the best averaged models have very large averaging windows, corresponding to slow decays with $\alpha \to 1$. It is important to note that we use a sampling of $T = 16$, which affects the effective decay rate (see Appendix F).

In Table 24 we report the best accuracy for each decay when training on CIFAR-100 with a ResNet-18. This corresponds to Fig. 1b, which we include again here (Fig. 9) for the convenience of the reader.

The main takeaway from Table 24 is that the slower the decay, the later it reaches its peak performance and the higher the accuracy is. However, note that this includes Batch Norm recomputation after every epoch; if Batch Norm is not recomputed the best decay is faster. In Section 3.4 we discussed this phenomenon: the model weights are more robust to large averaging windows than BN statistics.

| Decay rate $\alpha$ | Best test accuracy | Epoch |
|---|---|---|
| 0.968 | $78.04 \pm 0.19$ | 137 |
| 0.984 | $78.42 \pm 0.22$ | 140 |
| 0.992 | $78.81 \pm 0.16$ | 152 |
| 0.996 | $79.02 \pm 0.19$ | 156 |
| 0.998 | $79.06 \pm 0.14$ | 166 |

Table 24: Best accuracy and epoch when it was reached for the EMA models with BN recomputation plotted in Fig. 9.

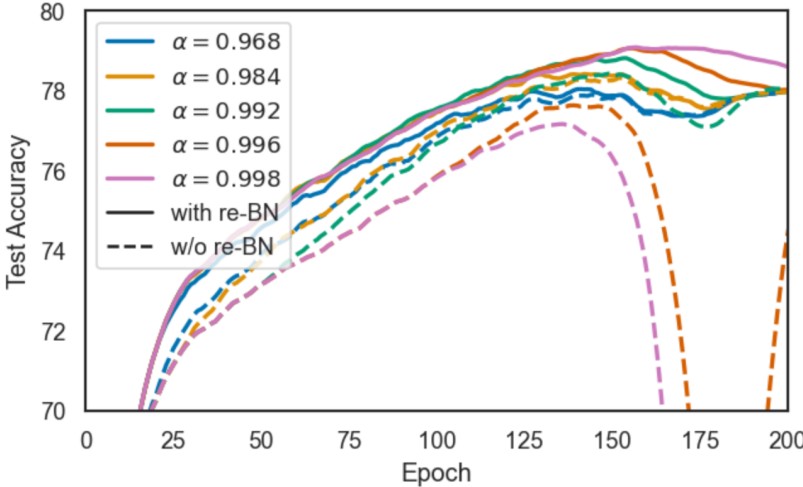

Figure 9: Breakdown of the 5 EMA models per decay (with and without BN recomputation after every epoch). EMAs with the largest averaging windows fail unless BN stats are recomputed. Sliding window of 5 used for smoothing. All results are the mean of 3 runs.

