# OpenReview forum: "Exponential Moving Average of Weights in Deep Learning: Dynamics and Benefits"
_TMLR — Accepted by TMLR_

### Review · Reviewer_CZGX · 2023-12-12

**Summary Of Contributions:**

- Weight averaging of Stochastic Gradient Descent (SGD) iterates is a widely employed technique in the training of deep learning models. Although it is frequently integrated into intricate training pipelines to enhance generalization or function as a 'teacher' model, the properties of weight averaging itself have not been adequately assessed. The authors tried to address this issue by conducting a numerical experiments mainly focusing on the Exponential Moving Average (EMA) of weights.
- The investigation in this paper begins by delving into the learning dynamics of EMA, providing suggestions for tuning hyperparameters, and partially clarifying its effectiveness as a teacher model and its properties in the early learning phase. The authors mentioned that these results claim to reveal that EMA can have a smaller learning rate decay rate than SGD because EMA inherently reduces noise and introduces a form of implicit regularization.
- Through an extensive array of experiments, the authors reported that EMA solutions diverge from the last-iterate solutions. Furthermore, they claimed that EMA models not only exhibit superior generalization but also manifest enhanced i) robustness to noisy labels, ii) **repeatability (?)**, iii) calibration, and iv) transfer learning capabilities.
- They concluded that incorporating an EMA of weights represents a straightforward yet potent augmentation to enhance the performance of deep learning models.

**Audience:**

Yes

**Broader Impact Concerns:**

I believe that this work does not raise any ethical concerns because it is an fundamental & empirical study focused on EMA with standard benchmark datasets in the machine learning context.

**Claims And Evidence:**

No

**Requested Changes:**

To rectify the earlier identified shortcomings, I deem it essential to provide clarifications and corrections to the summarized elements below. If there are any misinterpretations on my part, I would value your guidance in pinpointing them.

- Is it adequate to rely on a comparison only with the baseline to assert that EMA exhibits properties such as high generalization performance, robustness to label noise, good calibration, etc., particularly in the experiments outlined in Section 4? Put differently, what leads you to believe that the current numerical evaluation is comprehensive enough to address the inquiry, "What are the properties of weight averaging when training deep neural networks?” Please explain your reasons for this in a way that is acceptable to all reviewers. If you have no particular reason, please at least report the position of EMA based on your argument through more multifaceted numerical experiments, such as comparative experiments with SWA.
Otherwise, it is difficult, at least for me, to recommend the acceptance of this paper. This aspect is closely related to one of the main acceptance criteria of TMLR: *a discussion based on accurate evidence.*

- The authors asserted that EMA reduces noise; however, there appears to be no supporting experiment to validate this claim. Confirming this property through experimentation is crucial, as it is one of the significant aspects that should be substantiated.

- Please provide a rationale for the experimental setup, especially for the hyperparameter settings (e.g., number of bins) and their candidate values (e.g., $\alpha$ determined by validation, candidate learning rates). You may cite references if necessary, and if you have any ideas to eliminate arbitrariness, please state them clearly. This will be useful not only for readers to reproduce the results of the experiments in this paper, but also for designing the experimental setup. This aspect is also closely related to one of the main acceptance criteria of TMLR: *a discussion based on accurate evidence.*

- I am apprehensive about the experimental results of the comparison methods being exclusively quoted values. While this would be acceptable if the experimental designs, encompassing model structure, initial values, and all other pertinent settings, were identical across studies, it appears unlikely to be the case. If you believe the existing information is satisfactory, please provide a compelling rationale for this choice. In the absence of a specific justification, we recommend that the authors conduct replicated experiments themselves, addressing the performance to conduct a more fair comparison.

- The Appendix reports the mean ± std. of the experimental results, but the main paper reports only the mean values of the experimental values. If there is no particular reason, then the discussion should be based on the mean ± std.

- Please clarify the paper's position employing relevant figures and tables where applicable. It would be better to summarize the findings from pertinent studies and the presented facts in this paper, minimizing redundant explanations. Acknowledging the reader's experience, it's noted that understanding the content up to Section 3 posed challenges.

- The current version lacks sufficient descriptions of the EMA and SMA algorithms. A concise introduction of preliminaries for Student-Teacher setting, EMA, and SMA would greatly benefit novice learners.

- Please eliminate ambiguity by providing clear explanations or references that define crucial concepts, such as **repeatability**. Addressing this issue is pivotal for improving overall readability.

- Please provide explanations for any blank columns in the tables in the Appendix. If there is no specific reason for leaving them blank, please fill in the information.

- Please include the relevant references for the methods employed in hyperparameter selection, and conduct a thorough review for any typos in the manuscript.

- Ultimately, it is essential to clarify the core message of this paper. A well-organized section explicitly addressing the authors' question, "what are the properties of weight averaging when training deep neural networks?" would significantly enhance readability and provide a clear takeaway for the readers.

**Strengths And Weaknesses:**

First and foremost, I would like to express my sincere respect for all the efforts the authors have invested in this paper.
Furthermore, it should be noted that the following review is explicitly written with a clear understanding that TMLR places importance on *accuracy, convincing, and clear evidence,* as well as *capturing readers' interest* over novelty and impact.

## Strengths
- The authors have made an efforts to offer a comprehensive analysis of the diverse properties achieved through the utilization of EMA, relying on empirical evaluation. This approach represents an intriguing initial step in comprehending the effectiveness of EMA.

## Weakness
- **Concern for empirical evidences to validate the authors' claim:**
  - The current experimental results could raise concern about whether they are sufficient to assert the various properties of EMA in achieving its stated objective of elucidating each characteristic. The authors consistently employ SGD (with momentum) as the baseline throughout their study. However, it remains questionable whether using this method alone is enough to attribute the high generalization performance, robustness, and calibration capabilities to the specific features of EMA. The current results seem to indicate only an empirical advantage over the baseline rather than providing a solid basis for claiming unique characteristics of EMA, e.g., high generalization, robustness, and calibration. Moreover, the exclusive focus on image classification tasks in all experiments raises doubts about obtaining a comprehensive understanding of EMA's characteristics solely based on results from this singular experimental setting. A fundamental means to strengthen this claim would be to conduct experiments involving a broader range of tasks, e.g., regression. While the authors have conducted experiments involving transfer learning, the diversity in this aspect is not yet sufficient in the current version.
  - As you also mentioned in the introduction, SWA is a highly popular averaging method, garnering significant attention from researchers in the field of machine learning. Yet, why is there a lack of comparison with SWA in all experiments? I understand that in some cases, it might be challenging to adopt EMA as a teacher model due to the necessity of calculating repetitive batch normalization values. However, it seems that most of the current experimental settings are computationally feasible, and thus I believe there is no reason not to conduct a comparison with SWA. To comprehend the properties of EMA without theoretical analysis, characterizing EMA's performance in terms of robustness, generalization, calibration, etc., without systematically sorting out its advantages and disadvantages compared to recent popular methods like SWA, appears premature. Additionally, and this is a personal opinion, I suspect that many readers would find a comparison with SWA more intriguing than an assessment of how EMA performs compared to SGD baseline.
  - Furthermore, the authors have omitted most of the numerical values for the comparison experiments, relying on direct quotations from [Izmailov et al. (2018)] and the other papers summarized at the web site. For a fair and aligned comparison under consistent experimental settings, it is imperative to perform a replicated implementation and directly compare the performance.
  - The intention behind reporting only the mean on the main page, despite focusing on the standard deviation in the Appendix, remains unclear. Additionally, the experimental results cited in Table 2 (Wei et al., (2022); http:// www.yliuu.com/web-cifarN/Leaderboard.html) include standard deviation, but it is conspicuously absent here. This omission seems peculiar, and the reason for it is not apparent, giving the impression of a highly arbitrary presentation.
  - The selection of hyperparameters using validation data appears reasonable, but the choice of candidates, such as $[[0.968, 0.984, 0.992, 0.996, 0.998]]$ for $\alpha$, seems arbitrary. The progression appears to decrease by $(2*k)/100 \quad (k=1,2,3,4)$ from right to left, but the rationale for this is not clear. How was the validation range determined, and is there sensitivity analysis regarding how EMA performs with different $\alpha$ values? It's unfortunate that there are no experimental reports based on a variety of $\alpha$ settings, especially since some readers are likely interested in understanding the sensitivity to $\alpha$. Additionally, in Appendix B, the LR row for SGD is occasionally left blank, leaving the tuning of the baseline SGD learning rate unclear. The setting of $M=100$ for the Bin number in the ESE calculation lacks a specific explanation. If these arbitrary elements remain unjustified, they could significantly impact the validity of interpreting the experimental results presented in this paper.

- **Concern for the standpoint of this paper:**
  - While Sections 1 and 2 offer detailed reviews of related studies, it is challenging to discern the paper's unique contributions and where it stands in comparison to existing research. A helpful clarification, for example, would be to provide a table outlining the distinctions between the related studies and the current one, facilitating a clearer understanding of the novel aspects and contributions of this study.

## Concerns regarding the interest of TMLR readers
- To capture the readers' interest, readability is one of the crucial points. However, in this paper, it appears that the following elements are having a negative impact on readability.

  - **Some ambiguous words**: The presentation uses terms that may not be widely familiar (at least to me), creating readability issues. For instance, what is meant by "repeatability," and how does it differ from "reproducibility"? If these concepts are established elsewhere, please provide citations. If these are newly defined concepts in this paper, it would be preferable to offer a more detailed explanation. Additionally, the abrupt introduction of "averaging windows" just before Section 3 makes it challenging to understand without reading through Section 3.1. Such ambiguous expressions and the introduction of terminology without prior definition can be stressful for readers and may diminish interest. Clarifying these aspects would greatly enhance the clarity and accessibility of the paper.
  - **Inadequate information of algorithmic details**: This paper lacks comprehensive information on the algorithms it addresses. Considering the broad readership of TMLR, a precise comprehension of the paper's content is challenging without elucidations on concepts such as learning in the Student-Teacher setting, the utilization of EMA and SMA, among others. Without these explanations, a precise understanding of the paper's contents becomes challenging.
  - **Typos**: In the caption of Figures 1. and 4.: Learning rate $\mu$ --> $\eta$?
  - **Absence of Essential Citations**: Despite being a standard feature in several frameworks such as PyTorch, I believe it is necessary to include a reference, with [Loshchilov et al., 2021] being a suitable choice.

**Citation**:
- [Loshchilov et al., 2021]: Ilya Loshchilov and Frank Hutter. SGDR: Stochastic Gradient Descent with Warm Restarts. ICLR2021.

---

> ### Author Response · Authors · 2024-01-23
> **Authors' Answer to Reviewer CZGX [1/2]**
>
> Thank you for the careful review of our paper. Below we address the weaknesses (W) and requested changes (RC) point by point:
>
> **W1: concerns on experimental results**
>
> Needless to say, in an experimental paper, the more experimental results the better. In our approach we have chosen to focus on image classification benchmarks, and to provide a breadth of experiments on different architectures, datasets and properties of the EMA models to be tested. In order to realize this amount of experiments, we have limited the scope to image classification, a very mature task where benchmarks are curated, reliable and accepted (e.g., existing competitive benchmark for learning with label noise). Ever since ImageNet was introduced, image classification benchmarks have driven immense progress in the field of deep learning. While lately other vision and language tasks have gained a lot of traction, we believe this is not an argument to dismiss results from image classification-based empirical studies.
>
> About the baseline method used, SGD with Nesterov momentum is a competitive and widely used algorithm. We deliberately wanted to adapt an algorithm that is used in practice to show how applying EMA on top can readily improve the results of practitioners’ pipelines.
>
> **W2 and W3: comparison to SWA**
>
> We regard SWA and EMA as fundamentally similar. Both are averaging points along the SGD trajectory, simply with different weighting (SWA uniform, EMA weighting more the latest iterations). For this reason, and in light of the absence of such study in the literature, we believe it’s more relevant to study “Averaging weights vs not” than comparing two different ways of averaging weights. Between SWA and EMA, we focus on EMA since it’s more widely used in practice (particularly as teacher model) and because studying its training dynamics are of more interest (SWA is only used as final model, and not used online as a teacher, there is no such thing as its training dynamics).
>
> Nonetheless, it is very reasonable to expect EMA vs SWA results. Following the requested change, we have reproduced the SWA implementation and run the experiments on generalization, arguably the most important property. For that, we use the exact same experimental setup as for EMA. This experiments solve two issues of our previous version of the paper, the lack of a full comparison to SWA in Table 1 and the lack of replication of SWA results.
>
> **W4: missing standard deviation**
>
> Following the requested change, we have added the standard deviation in Table 1, 2, 3 and 5. The original reason to omit standard deviation in the main text and feature it only in the appendix was to improve readability. The standard deviation was always small enough to not affect the conclusions of the results. We still added it in the appendix for completeness and soundness of the experimental study, but decided to omit it in the main text in order to improve the readability and clarity of the conclusions.
>
> **W5: hyperparameter selection**
> - To provide more clarity on the effect of the decay rate, we have included Appendix G, where we report a sensitivity analysis on the EMA decay $\alpha$.
> - The choice of candidates for decay $\alpha$ approaches asymptotically the limit of 1. From previous EMA literature and experiments, the best values correspond to the smallest decays, when alpha approaches 1. Exploring how close we can get to 1 by approaching asymptotically was the most sensible approach, thus the exponential range of decays instead of linear. We have added a clarification on this in Appendix G.
> - Thank you for pointing out that the selection of LR for SGD was unclear in the tables of the appendix, following this concern, we have updated them. As studied in Appendix D, the optimal learning rate for SGD and for EMA actually show to be aligned.
> - The selection of M=100 bins is admittedly somewhat arbitrary, as some number of bins needs to be chosen. This selection follows previous literature in other calibration methods.
>
> **W6: standpoint of the paper**
>
> We have updated the conclusions to better highlight the unique contributions of the paper, facilitating a clearer understanding of the insights gained from our study.
>
> **W7: Ambiguous words**
>
> Thanks for expressing this concern, in order to improve the clarity of the text we have removed the word “repeatability” and referred to it consistently as “prediction consistency”, which is properly defined with a detailed explanation in Section 4.4

---

> ### Author Response · Authors · 2024-01-23
> **Authors' Answer to Reviewer CZGX [2/2]**
>
> **W8: Algorithmic details**
>
> The EMA algorithm is very simple: we keep a moving average of weights, as defined in Eq. 1 and evaluate on it. We estimate Eq. 1 to be clear and to fully capture the EMA algorithm.
>
> As for the teacher-student framework, this algorithm is not part of the paper, nor is it used anywhere in the paper - it is only a class of algorithms that make use of EMA. Thus, we don’t estimate relevant to include it, and refer the interested reader to the cited literature on this topic. Similarly for SWA, which we don’t use in our paper and refer to the original work for details.
>
> **W9: Typos**.
> Thank you, we have corrected the caption of Fig. 1 to use consistently $\eta$ for LR.
>
> **Citation missing:** Thank you, we have included the proposed citation.
>
> **RC 1: Comparison to SGD baseline is not convincing**
>
> In a randomized controlled trial, the aim is to compare two populations under the same conditions and change only one variable to evaluate its effect. Since the main question of the paper is "What are the properties of weight averaging when training deep neural networks?”, the most natural baseline to compare with is the *exact* same training trajectory without weight averaging. In our experimental setup, we compare literally the same training run (thus controlling even for the factor of stochasticity in deep learning optimization) and change only one variable: to average or not to average. We believe this is the most effective and convincing way to evaluate the impact of averaging, as the great differences we observe between EMA and SGD trajectories can only be due to and explained by averaging. We hope that this explanation has been clear enough to justify the design of our experiments, we remain open for discussion if the reviewer has any further comments.
>
> With this approach, we are able to identify multiple differences in both dynamics during training (Section 3) and properties of the final solution (Section 4) between averaged and non-averaged models, providing an overview of their differences in multiple aspects relevant for practitioners. Additionally, for various properties (Sections 4.3, 4.3, 4.6) we compare the EMA results against specialized methods. To improve the comprehensiveness of the study we have also included additional SWA results to compare EMA to, providing a better picture of weight averaging in different forms.
>
> **RC 2: Supporting evidence that EMA reduces noise**
>
> Weight averaging is known to reduce noise of stochastic updates in convex optimization, in particular for quadratic objectives [Polyak & Juditsky, 1992]. For general non-convex objectives, such clear theoretical results cannot be obtained, and one can only rely on empirical results. Our results show clearly a benefit of using EMA compared to the baseline SGD (see how EMA dominates SGD entirely in Fig. 1a, for example), the reason being, similarly to the convex case, noise reduction.
>
> While we do not have specific experiments to test noise reduction, we believe that this aspect is embedded in many of our current experiments. As a matter of fact:
> - The benefits of EMA over the baseline fade when the step-size diminishes. This is because there is less variance in the baseline sequence then, and so little is gained from diminishing it.
> - A natural idea is to use EMA in the optimization algorithm to leverage its effectiveness. Unfortunately, as described in Section 3.3 and Appendix C, performing gradient descent from the EMA iterates ($x_{t+1} = x_t^{EMA} - \eta g_t$) does not improve the performances, and the good performance of EMA iterate is destroyed in only one step. This is because the update direction $g_t$ is noisy (even when using momentum), and applying it only one time is enough to reinject noise in the EMA estimator and induce a performance collapse. This would certainly not be the case if the performance of EMA was not due to variance reduction.
>
> **RC 3**: Addressed in answer to W5.
>
> **RC 4**: Addressed in answer to W3.
>
> **RC 5**: Addressed in answer to W3.
>
> **RC 6 and RC 10: clarify contributions**
>
> Following these comments, we have fully re-written the conclusions section to clarify the core contributions of the paper and the main takeaways.
>
> **RC 7**: Addressed in answer to W8.
>
> **RC 8**: Thank you for the concern raised, we have filled the remaining blank values of tables in the appendix.
>
> **RC 9**: Addressed in answer to W5.
>
>
> We remain available for further comments and discussion

---

> > ### Comment · Reviewer_CZGX · 2024-02-06
> > **Acknowledgements and Comments to the Authors' Response**
> >
> > Thank you for your careful reading my review and insightful responses, and I apologize to my late reply.
> > I summarize my comments as follows.
> >
> > ## W1: concerns on experimental results
> > - As you rightly pointed out, numerical experiments in image classification benchmarks have been instrumental in driving significant progress in the field of deep learning, and I fully agree with the choice of experimental settings to discuss the nature of methods. What I intended to convey is that the properties of EMA and SWA elucidated in this paper should be explicitly stated as being limited to the specific problem setting of image classification tasks, without intending to negate the adoption of numerical experiments based on such tasks altogether. In other words, in my opinion, it should be stated clearly that EMA does not guarantee its properties in regression or other vision and language tasks as a limitation. Considering this, it might be beneficial to include a ``Limitation'' section, incorporating the aforementioned points, to further strengthen and enhance the credibility of this paper.
> >
> > ## W2 and W3: comparison to SWA
> > - Thank you for adding the experimental results of EMA and SWA. I believe that it has enhanced the impact of this paper from the perspective of TMLR's acceptance criteria: ``capturing readers' interest.''
> >
> > ## W4-W7, W9
> > - Thank you for revising it carefully. With this modification, I believe that weaknesses regarding this aspect have been alleviated to some extent.
> >
> > ## W8:
> > - It seems a good idea that providing references to the original papers or relevant literature on the algorithms of EMA and SWA. The algorithms of EMA and SWA may seem simplistic as you said. However, considering one of the examples of acceptance criteria in TMLR, which includes educational value to students (see [https://jmlr.org/tmlr/acceptance-criteria.html]), I personally think it would be beneficial to introduce them. (Of course, this is just a personal opinion, and I believe it would suffice if there is an indication or reference to the original papers of EMA and SWA as mentioned above.)
> >
> > ## RC1--RC9:
> > - I confirm that my revision requests have been largely fulfilled. I would like to express my gratitude for the careful revisions and convey my respect for the efforts of the authors to enhance the quality of this paper.

---

> > > ### Author Response · Authors · 2024-02-08
> > > **Thank you for your response**
> > >
> > > Thank you for the response and continuous feedback, which is definitely helping to improve the paper overall.
> > >
> > > We are delighted to see that most of the unclear points and requested changes are now largely addressed. We take the points regarding explicitly stating the limitation of experiments to image classification, and the educational perspective of the paper. We will work on improving those in the final version of the text.

---

### Review · Reviewer_G5bH · 2023-12-25

**Summary Of Contributions:**

This study focuses on the properties of Exponential Moving Average (EMA) of weights in neural networks. The authors investigate the training dynamics of EMA when used in combination with the popular Stochastic Gradient Descent (SGD) optimizer. Through a series of experiments, they demonstrate various properties of EMA, such as good early performance, robustness to label noise, repeatability, better calibration, and transferability. Moreover, they also reveal that EMA requires less weight decay to achieve good performance since it introduces a form of implicit regularization.

**Audience:**

Yes

**Broader Impact Concerns:**

No specific concerns

**Claims And Evidence:**

Yes

**Requested Changes:**

While the authors evaluate the effect of EMA using a large dataset, CIFAR100, and several architectures, I still think some systematic evaluation of the properties of EMA is lacking. I think it would be valuable to have a controlled environment with synthetic data to demonstrate the effect of EMA on generalization, calibration, and robustness to label noise.
Since the authors use several EMAs (M=5) there is a computational overhead compared to the baseline. This is not a big issue, but in terms of generalization, it provides some conceptual advantage for the EMA compared to the baseline. This is because the baseline is based on one model and not five (I know that this is one run, but the prediction is still based on the best of five versions). One alternative would be to compare to an ensemble of perturbed weights, for example, using one of the following schemes:

[1] Sharma et al. Investigating Weight-Perturbed Deep Neural Networks With Application in Iris Presentation Attack Detection.
[2] Khatami et al. A weight perturbation-based regularisation technique for convolutional neural networks and the application in medical imaging.

Why are you using momentum? Momentum also has a moving average of the gradient, so it seems more reasonable to use vanilla SGD to decouple the effect of the EMA from the effect of momentum.

Some minor comments:

Above eq. 1 x_t isn’t in the same font as in the equation and isn’t bold

You claim that SGD:

“While SGD tends to land near a sharp ascent”

And then

“Noisy SGD updates are argued to bias solutions towards flatter regions that are believed to generalize better, partly explaining the success of deep learning (Keskar et al., 2016).”

These statements are somewhat contradict each other.

In the caption of Figure 1, the learning rate should be \eta and not \mu
Also, it is not really clear from the caption how the left figure was created; this should be explained. Specifically, how you chose the best value of \alpha for each epoch in the training.

Why are some values missing in Tables 2 and 3?

In Table 3, the comparison to Co-distillation KL is somewhat misleading since both were not conducted using the same architecture.

Section 4.5- it is not clear from which data to which data was the transfer learning conducted.

It would be informative to show visually (by showing the loss and acc) how the two different early stopping criteria behave across datasets.

**Strengths And Weaknesses:**

Strengths: The authors have tackled an important issue in Machine Learning: the impact of Exponential Moving Averages (EMA) on Neural Networks (NN) performance. This problem has various implications and is of interest to the research community. The paper's English level is good overall.

The authors have used multiple datasets to demonstrate the various properties of EMA, all of which show the benefits of EMA compared to vanilla Stochastic Gradient Descent (SGD).

Weaknesses: Although the paper does not provide any theoretical results or explanations for the findings, it is still an experimental paper, and I appreciate that. However, I believe that even as an experimental paper, there should be a more systematic evaluation to provide a better understanding of the effect of EMA on the properties of NNs.

Furthermore, some aspects of the paper's presentation need improvement to enhance its readability. I have elaborated on these issues in the following section.

---

> ### Author Response · Authors · 2024-01-23
> **Authors' Answer to Reviewer G5bH [1/2]**
>
> Thank you for the careful review of our paper! We are happy to see that the topic is considered to be of interest to the community and that the experimental results are appreciated. We address the requested changes point by point below.
>
> **Requested Change 1: on systematic evaluation and experimental setup**
>
> As an experimental study, we have made an effort to define a sound, comprehensive and reproducible experimental set up that provides a systematic evaluation of EMA models. For that, we report experiments on a breadth of datasets (CIFAR-10, CIFAR-100, TinyImagenet) and architectures (ResNet-18, WideResNet-28-10, VGG-16) and reach consistent conclusions across all of them. These are all well-known datasets and architectures with competitive benchmarks which have driven great progress in deep learning during the past decade, for which we consider reliable to evaluate on them. We estimate that using these familiar benchmarks is more convincing than designing a controlled environment with a synthetic dataset, which could potentially raise questions on its generation and design choices.
>
> To perform a rigorous study, we tune hyperparametes on a hold-out set of data, which is unfortunately not the common practice in deep learning research. Our procedure (defined in Section 4.1 and Appendix F) is to split the training set in train/validation (80/20 split) and tune the hyperparameters on the validation set. Then, to compare to works in literature, we use the entire train set, but do not further tune any hyperparameter on the test set, instead we use the values tuned beforehand on the validation set.
>
> Regarding the use of multiple EMA decays, we do not agree that it provides a conceptual advantage to the baseline. Using an EMA introduces one hyperparameter, the decay rate $\alpha$. The reason to use $M=5$ models is to tune $\alpha$. We propose a one-shot tuning procedure (a single training run is required) which consists of keeping M parallel EMA models, which is feasible since the overhead is orders of magnitude lower than the computational and memory requirements of the optimization. We only use the M models to choose the best $\alpha$ **on the validation set**, as explained in the previous point. Then, when training on the entire train set, we **only use one EMA model**, with the tuned $\alpha$. The tuned decay rates for each experiment are reported in the tables of Appendix B. It is true, nonetheless, that we use the M EMA models in the plot of Figure 1a (reporting the best out of the M models plotted in Figure 1b), we have updated the text to clarify any possible confusion. In all results reported in Tables, the accuracy corresponds to a single EMA model with decay rate tuned on the validation set - we have also updated the text to try to clarify this point.
>
> **Requested Change 2: On using momentum**
>
> We use SGD with momentum as training algorithm to provide the most realistic and competitive baseline. The use of momentum in SGD training is widespread and has been repeatedly shown to improve performance. While it introduces an averaging of the gradients, it is conceptually different from an EMA. Momentum averages gradients, which has an implication on the direction of the next update, providing a more stable trajectory where consecutive updates move generally in the same direction. EMA, on the contrary, averages weights, that is, points along said trajectory. Another difference is that momentum is part of the optimization algorithm, while EMA is completely offline, applied only on top of the underlying training trajectory. Therefore, EMA is complementary to momentum as it provides further variance reduction, that would slow the algorithm down if it was ensured only through momentum: iterates continue to progress fast (and with some noise, but in a more stable way thanks to momentum), but we evaluate the error at the EMA of these iterates, which is far less noisy.
>
> Finally, our goal is to study EMA on top of the most widespread and competitive training algorithm. Therefore, we believe that not using momentum would raise questions on the quality of the baseline and the conclusions from the experimental results.
>
> **RC 3 and 5: Typos**
> Thank you, we fixed the font of $x_t$ and $\mu$ to $\eta$ in Figure 1.

---

> ### Author Response · Authors · 2024-01-23
> **Authors' Answer to Reviewer G5bH [2/2]**
>
> **Requested Change 4: On contradictory statements from SGD literature**
>
> The statements cited from SGD literature, from influential works in the field, are attempts to explain a notoriously complex phenomena, the success of SGD to optimize non-convex functions. While they might appear contradictory at first, they are actually compatible. [Keskar et al, 2016] argues that the neighborhood where SGD finds a solution is biased to be flatter. It does not argue that it is absolutely flat in all directions, which would be particularly difficult considering the high dimensionality of the problems. For those dimensions where the neighborhood is not flat, [He et al, 2019] shows there is often an asymmetric valley: a sharp ascent in one direction and a flatter surface on the other. This observation on the geometry of the loss provides a plausible explanation for the success of averaging weights in deep learning. We refer the reviewer to He et al [2019], for a more detailed and very interesting discussion on this topic.
>
> **RC 6: Missing values in Tables 2 and 3**
>
> Table 2 shows the top methods on the leaderboard from http://www.yliuu.com/web-cifarN/Leaderboard.html. Following the requested change, we have extended the list of methods that we feature, Nonetheless, we decide not to include the entire leaderboard for readability purposes and omit the lesser relevant methods, referring the reader to the source instead for full details.
>
> Table 3 compares our EMA results to the SGD baseline and to the one work in the literature that proposes a method to reduce prediction disagreement, [Bhojanapalli et al., 2021]. Unfortunately, the authors did not report results using JS divergence and do not provide a source code or clear reproducibility instructions, for which JS results are not available for their method.
>
> **RC 7: clarify datasets for transfer learning**
>
> Following the requested change, we have updated Table 4 and the text in Section 4.5 to clarify which dataset was the model pretrained on and on which dataset do we evaluate. Please let us know if this would need any further clarification.
>
>
> We remain available for further comments and discussion.

---

> > ### Comment · Reviewer_G5bH · 2024-02-10
> > **Response to authos**
> >
> > I thank the authors for their response and changes in the paper. Overall, my questions and comments have been addressed.
> >
> > My only minor concern that remains is the comparison to a simple model with 5 perturbed weights and selecting the best one or ensembling using one of these schemes:
> >
> > [1] Sharma et al. Investigating Weight-Perturbed Deep Neural Networks With Application in Iris Presentation Attack Detection. [2] Khatami et al. A weight perturbation-based regularisation technique for convolutional neural networks and the application in medical imaging.

---

### Review · Reviewer_UxSd · 2024-01-15

**Summary Of Contributions:**

This paper presents a detailed investigation into the effectiveness of Exponential Moving Average (EMA) weight averaging in deep learning. Through extensive empirical studies, it uncovers that:

1. EMA models enable the use of higher learning rates. The authors argue that this leads to an implicit regularization effect that promotes the learning of better representations.

2. EMA models have notable early performance. The paper also cautions against overly large averaging windows due to the need for recalculating Batch Norm statistics.

3. EMA models have several significant benefits on top of SGD iterates. Beyond improved generalization that has been previously acknowledged in the literature, EMA models also exhibit enhanced robustness to label noise, increased consistency in predictions, the ability to produce more versatile and transferable representations, and better calibration.

These insights suggest that incorporating EMA into deep learning models is a simple yet potent strategy to enhance performance across various dimensions.

**Audience:**

Yes

**Broader Impact Concerns:**

No ethical concerns.

**Claims And Evidence:**

Yes

**Requested Changes:**

This paper may consider the following improvements:

Essential to my recommendation:
- A more detailed explanation or additional empirical data is needed to clarify the early-stage benefits of EMA, specifically addressing the observations made in weakness 2.
- A sensitivity analysis focusing on lr adjustments is crucial, as indicated in weakness 3.

Secondary consideration, suitable for subsequent research:
- Although less pressing, a comparative analysis with other techniques addressing similar goals, as mentioned in weakness 1, would provide valuable insights.

**Strengths And Weaknesses:**

**Strength**:

This paper is the first work to provide a detailed and specific analysis of the benefits and limitations of the EMA. It proposes several benefits of EMA such as improved generalization, exceptional early performance, robustness to label noise, consistency in predictions, etc. The finding on early performance is particularly insightful, explaining the success of EMA in the teacher-student setting. Robustness to label noise and consistency in predictions are novel findings, previously unaddressed in the literature. Furthermore, the paper is well-written. Key points are articulated clearly and concisely, making them accessible and informative.

**Weakness**:
1. Experiments are confined to image classification. Image classification is a relatively mature task in the field and has many design choices. While EMA offers improvements in generalization, prediction consistency, and so on, the paper does not compare the advantages and disadvantages of EMA with other techniques aimed at achieving similar goals. This limits the understanding of EMA's unique benefits over other existing methods.

2. The paper repeatedly claims that a key benefit of EMA is its ability to "reduce noise and potentially replace the last phase of learning rate decay" while enabling the implicit bias of SGD. However, this claim is somewhat confusing to. If EMA's primary role is to save on the last iterations of SGD, why does Figure 1(a) show that EMA significantly outperforms SGD in the very early stages of training (e.g., the first 50 epochs)? Shouldn't the implicit bias of stochastic noise also be effective in the early phases of SGD?

3. The experiments suggest that the effectiveness of EMA may require a meticulous learning rate (lr) schedule, such as choosing an optimal initial learning rate after applying cosine annealing. It is unclear how sensitive EMA is to lr changes and whether fine-tuning the lr for SGD alone could also lead to similar improvements. This uncertainty is a notable gap in the paper's analysis.

---

> ### Author Response · Authors · 2024-01-23
> **Authors' Answer to Reviewer UxSd**
>
> Thank you for the careful review of our paper! We are glad to see that the topic was found to be relevant and the results insightful. Below we address the weaknesses and requested changes point by point:
>
> **Weakness 1 (and Requested Change 3): Experiments are confined to image classification, a mature task. Missing comparisons to other techniques aimed at achieving the same goals.**
>
> Image classification is indeed a mature task, and a reliable benchmark that has driven great progress in deep learning research in the last decade. We choose to focus on this task and test multiple properties of the models on a breadth of datasets and architectures.
>
> Regarding the comparison to other methods, we believe that we are comparing to relevant techniques for every property. For generalization, the baseline of momentum SGD is competitive and has been largely optimized and reproduced, and we also compare it to SWA, for which we have included more results in the last submission. For robustness to label noise we compare our model in the leaderboard of the CIFAR-N benchmark. For calibration we compare to the popular Temperature Scaling calibration method. Finally, for prediction disagreement we compare to the existing literature. Could the reviewer be more specific about which other techniques we are missing a comparison to?
>
> Finally, we would like to point out that the most important comparison is always against the SGD baseline. Our aim is to assess the impact and properties of weight averaging, and the best way to do that is by comparing EMA against the *exact* same training run without averaging (exact same setting, controlling even for the factor of stochasticity in training).
>
> **Weakness 2 (and Requested Change 1): Shouldn't the implicit bias of stochastic noise also be effective in the early phases of SGD?**
>
> Yes, it is. The noise reduction effect of EMA is effective all throughout training. We distinguish between the effects on early and later stages of training because we estimate that they have different and complementary implications for deep learning practitioners.
>
> The strong early performance of EMA is an interesting insight that sheds light on the effectiveness of EMA as a teacher model. Nonetheless, early EMA models still perform lower than a converged SGD performance. Practitioners are most interested in the best final performance, and less interested in the performance during the optimization. For this reason we emphasize the result of the later EMA improving the generalization of the best SGD model, while also reducing the number of training steps needed to achieve that performance.
>
> As per the requested changes, we have modified the text to clarify the point that noise reduction brings EMA a strong early performance. We dedicate Section 3.3 to highlight and discuss the early stage benefits of EMA, accompanied with experimental results on a breadth of datasets and architecture (in addition to Figure 1, see Appendix A). Furthermore, we find that the implicit regularization of EMA in the early stage is particularly useful in training with noisy labels, as discussed in Section 4.3. The strength of that implicit regularization is determined by the learning rate. To mitigate the impact of noisy labels, regularization should be stronger, and thus the early stage performance is actually better than the later stage. Finally, we have updated the conclusions section to improve the clarity on the results and takeaways, where we include the early stage benefit of EMA.
>
> If the reviewer still feels that early-stage benefits of EMA are not addressed with enough clarity, we will be happy to receive feedback on specific action points to improve.
>
> **Weakness 3 and Requested Change 2: Sensitivity to LR schedule and tuning**
>
> We agree with the reviewer that the learning rate is a crucial hyperparameter in training of deep models. This is the case when using EMA, just as it is the case for standard SGD training or for any other algorithmic technique. We would like to refer the reviewer to Appendix C, where we perform a sensitivity analysis to understand the impact of the initial LR on EMA models, which is the sensitivity analysis requested. In this section we show results and training curves of SGD and EMA models for different initial learning rate values. Interestingly, our results show that the optimal values for initial learning rate between SGD and EMA are aligned.
>
>
> We remain available for further comments and discussion.

---

> > ### Comment · Reviewer_UxSd · 2024-02-16
> >
> > I sincerely thank the authors for the response and corresponding modifications. My comments are properly addressed.

---

### Decision · Action_Editor_GP8B · 2024-03-06

**Recommendation:** Accept as is

**Comment:**

The validity of experimental results/design, lack of comparison with SWA, and readability were raised as primary concerns of the paper. The authors addressed almost all concerns well and adequately reflected them in the revision.
The findings obtained through extensive experiments are considered sufficiently generalizable. In light of the TMLR evaluation policy, the paper meets the acceptance criteria. However, as suggested by the reviewer, it should be clearly stated in the conclusion or limitation section that this finding is currently limited to image classification tasks.

**Audience:**

EMA is often used as a teacher model in semi-supervised learning and consistency training. Hence, this detailed empirical study will be of interest to the TMLR audience, particularly to the CV community.

**Claims And Evidence:**

This work studied the empirical nature of Exponential Moving Average (EMA) to see its effectiveness. Throughout the extensive experiments, the authors discovered interesting features of EMA: high generalization performance especially in the early phase of training, improved robustness to label noise thanks to high learning rate, and increasing consistency in predictions. These findings are supported by well-designed experiments.